# A Local Analysis of a Mathematical Pattern for Interactions between the Human Immune System and a Pathogenic Agent

**DOI:** 10.3390/e25101392

**Published:** 2023-09-28

**Authors:** Florian Munteanu

**Affiliations:** Department of Applied Mathematics, University of Craiova, Al. I. Cuza, 13, 200585 Craiova, Romania; florian.munteanu@edu.ucv.ro; Tel.: +40-723-529752

**Keywords:** local stability, dynamical systems, mathematical model for immune system–virus interaction, prey–predator model, system of ordinary differential equations, 37C75, 37N25, 37C20, 34C23, 34D20, 92C60, 92D30

## Abstract

In the present study, we introduce a four-dimensional deterministic mathematical pattern in order to study the interactions between the human immune system and a virus. The model is based on a system with four first-order ordinary differential equations, and the main aim of the paper is to perform a mathematical analysis of the local behavior of the associated dynamical system using the tools of the qualitative theory of dynamical systems. Moreover, two types of patterns with controls were introduced; consequently, some very interesting theoretical conclusions with medical relevance were obtained.

## 1. Introduction

In this paper, we consider a four-dimensional prey–predator model, and this is modified with three additional hypotheses in order to study the interaction between the immune system and a virus. The analysis is local and uses the tools of dynamical systems theory. The present study comprises a strictly theoretical aspect with limited applications in the study of the human immune system.

Structurally speaking, the human immune system can be divided into three components: organs, cells, and molecules [1,2]. According to the medical literature [1,2], the organs of the immune system are as follows: tonsils and adenoids, thymus, lymph nodes, spleen, Peyer’s patches, appendix, lymphatic vessels, and bone marrow. The cells of the immune system are as follows: lymphocytes (more exactly, T-lymphocytes, B-lymphocytes, plasma cells, and natural killer lymphocytes), monocytes, macrophages, and granulocytes (neutrophils, eosinophils, and basophils). The molecules of the immune system are represented by antibodies, complements, cytokines, interleukines, and interferons. The immune response of the human body comprises the collective and co-ordinated response to the presence of alien substances in an individual, and this response is mediated by the cells and molecules of the human immune system. The role of the immune system is to defend the body against microbes, to stop the growth of tumor cells by killing these cells, and to destroy abnormal or dead cells [1,2].

Generally, the immunity of a host body comprises the resistance against the invasion of pathogens and their toxic effects. When the presence of a pathogen agent is detected, any host body produces a nonspecific response (called innate immunity) and a specific response (called acquired immunity). Innate immunity relies on the mechanisms that exist before the infection and represents the first line of defense, but it has no memory for the next exposure and is based on nonspecific mechanisms. However, acquired immunity develops after the entry of pathogenic agents into the host and acts only after innate immunity fails to stop the invasion of viruses or microbes. This type of immunity has the memory to deal with subsequent exposure and acts using specific cells: T cells (mediated by cells) and B cells (mediated by antibodies) [1,2].

The immune system can be viewed as a system that is controlled by negative feedback. The central component of the system is represented by lymphatic tissues, which include mature T (thymic) lymphocytes that have matured via development in the thymus and mature B lymphocytes that have matured in the bone marrow. Generally, this essential system of the human body has a primary immune response, which is short-lasting and smaller in magnitude, and a secondary immune response, which is longer in duration and larger in magnitude, developing the so-called “cell’s memory” after the primary response. Failure of immune responses can result in hypersensitivity and immunodeficiency [1,2].

The main fighters of the immune system with respect to pathogenic intruders are the white blood cells, which move through blood and tissues throughout the body to attack invaders. The cells are created in the bone marrow and are part of the lymphatic system. There are five classes of white blood cells, namely neutrophils at 62%, eosinophils (acidophiles) at 2.3%, basophils at 0.4%, lymphocytes at 30%, and monocytes at 5.3% [1,2,3]. The typical lifetime of white blood cells varies from hours to days [4,5]. Because two classes of them, neutrophils and lymphocytes, represent about 92% of the total white blood cells, in [6], G. Moza and L.F. Vesa carried out a complete study of the “fight” of the immune system against the pathogen agents by restricting interactions with only the following two types of “bodyguards”: neutrophils and lymphocytes. This approach can be extended to three kinds of white blood cells (neutrophils, lymphocytes, and monocytes), which represent 97.3% of the total white blood cells, and this was carried out in a similar study [6], but it was more complex. Moreover, we can consider all five types of white blood cells, but the mathematical model of the resulting six-dimensional dynamical system will be very complicated because no less than 63 equilibrium points will appear [7].

The main aim of this paper is to construct a four-dimensional deterministic pattern to study the interaction between a virus and the human immune system, and the pattern is represented by the following three components: innate immunity, humoral immunity (after passing the infection or by vaccination), and cellular immunity (only after passing infection). This approach is new, although it is based on a modified predator–prey methodology used in population dynamics [8,9,10,11,12,13]. Some initial hypotheses used in classical prey–predator patterns need to be changed in order to take into account the types of interactions between the immune system and the virus (predator). Because a prey does not attack and kill a predator in classical predator–prey models [14,15,16], which are also known as Lotka–Volterra models, and the number of prey increases indefinitely in the absence of predators, we need to change these two premises to correspond to the reality of the interactions that we want to model. Thus, in our study, the two combatants are simultaneously predators and prey; in addition, the prey does not increase indefinitely in the absence of predators but stabilizes around a threshold. Several prey–predator patterns have been studied in [17] to investigate the relationship between host immunity and the growth of parasites. A pattern based on a system with four ordinary differential equations to model the interactions between an invading pathogen and the innate immune system, which is characterized by plasma cells, antibody concentrations, and a health factor, was presented in [18].

When taking into account that the immunity system has three components (innate immunity, humoral immunity, and cellular immunity), in the second section, we construct a four-dimensional deterministic model with three friendly species fighting the same combatant enemy—a pathogen agent. Knowing that, in reality, there are very complex connections between the three types of immunity, in order to obtain an appropriate mathematical model, we are forced to assume that these immunities act independently of each other and fight together to eliminate the pathogen from the body.

The local analysis of this theoretical model is carried out in the third section with mathematical tools from the dynamical system theory [19,20,21,22]. This type of system of ordinary differential equations is commonly used in mathematical patterns that model the dynamics of populations, the spread of diseases, or predator–prey interactions [23,24,25,26,27,28].

Next, in the fourth section, we present some very interesting medical interpretations about the behavior of the modeled system using 15 equilibrium points. Finally, in the fifth section, we introduce two types of medical control for the 4D system using three control functions in order to study the possibility of helping the human immune system in its fight against a virus. The obtained results suggest that by increasing at least one category of the immune system and using medical interventions in the early stages of the viral infection beyond the normal threshold, it is possible that the immune system will win the fight against the pathogenic agent despite low levels of other components.

## 2. The Methodology of the Construction of the Four-Dimensional Mathematical Model

In this section, we introduce a deterministic model for the interplay between the human immune system and a pathogenic agent. We use $x_{1}\left(t\right)$, $x_{2}\left(t\right)$, and $x_{3}\left(t\right)$ to denote the level of three types of immunity: innate immunity, humoral immunity, and cellular immunity. These three types of immunity fight together against a virus with the aim of stopping the virus’s propagation and eliminating it. The level of the viral load is denoted by v(t), and this is the number of viral cells that exist in the body at time *t*. We consider time as being continuous: t≥0. Thus, ${\dot{x}}_{1}\left(t\right)$, ${\dot{x}}_{2}\left(t\right)$, ${\dot{x}}_{3}\left(t\right)$, and $\dot{v}\left(t\right)$ represent the rates of change of these four quantities in a short period of time, ${\dot{x}}_{i}\left(t\right)=dx_{i}/dt$, $\dot{v}\left(t\right)=dv/dt$.

By taking into account the ideas from [6], we intend to construct a four-dimensional deterministic pattern for the interplay between the three types of immunity and a virus. Therefore, we assume that the following assertions hold.

**A1.** Without the presence of the virus, levels $x_{1}\left(t\right)$, $x_{2}\left(t\right)$, and $x_{3}\left(t\right)$ can be present in the body up to a certain threshold level. This assumption is based on the fact that any individual may have innate immunity and also humoral and cellular immunity before possible contact with a pathogenic agent. Thus, without any contact with virus v(t), we can consider that $x_{1}\left(t\right)$, $x_{2}\left(t\right)$, and $x_{3}\left(t\right)$ respect the following differential equations:$$\left\{\begin{array}{ccc} {\dot{x}}_{1} & = & a_{1}x_{1}-b_{1}x_{1}^{2}\\ {\dot{x}}_{2} & = & a_{2}x_{2}-b_{2}x_{2}^{2}\\ {\dot{x}}_{3} & = & a_{3}x_{3}-b_{3}x_{3}^{2} \end{array}\right.$$
where $a_{i}>0$ and $b_{i}>0$, i=1,2,3.

Let us remark that the general solution of the logistic equation ${\dot{x}}_{i}=a_{i}x_{i}-b_{i}x_{i}^{2}$, with the initial condition $x_{i}\left(0\right)=x_{i}^{0}$, is
$$x_{i}\left(t\right)=a_{i}x_{i}^{0}\frac{e^{ta_{i}}}{a_{i}+b_{i}x_{i}^{0}\left(e^{ta_{i}}-1\right)}\;,\;\;t\geq 0$$
and then we have $x_{i}\left(t\right)\to a_{i}/b_{i}$ for t→∞ and for any $x_{i}^{0}$, i=1,2,3.

If term $-b_{i}x_{i}^{2}$ is missing, then the general solution of the equation in $x_{i}$ will be $x_{i}\left(t\right)=x_{i}^{0}e^{ta_{i}}$, and of course, $x_{i}\left(t\right)$ increases exponentially when t→∞. Otherwise, if $b_{i}>0$, then for each i=1,2,3, the maximum value of the level of each type of immunity $x_{i}$ will be $a_{i}/b_{i}$.

**A2.** Usually, in a normal body without any autoimmune problems, the three types of immunity do not attack each other. Thus, these are destroyed only by the viruses’ actions; consequently, a term in the form of $-c_{i}x_{i}v$ will be added to each equation in $x_{i}\left(t\right)$, where v(t) is the level of viral cells that are in the human body at time *t*. Then, we obtain the next differential equation as an evolution law of $x_{i}\left(t\right)$:$${\dot{x}}_{i}=a_{i}x_{i}-b_{i}x_{i}^{2}-c_{i}x_{i}v\;.$$

**A3.** Without any immune response from the human body, the virus can multiply indefinitely and exponentially, which means that v(t) follows the differential equation $\dot{v}=p_{4}v$. Otherwise, if the immune system works properly, then the level of viral cells will decrease; consequently, we must add terms $-p_{i}x_{i}v$, i=1,2,3 to the differential equation of *v*. Consequently, this results in the next differential equation for the interaction between $x_{i}\left(t\right)$ and v(t), i.e., $\dot{v}=p_{i}v-p_{i}x_{i}v$.

Therefore, if we change the notation for the fourth variable, *v*, by $x_{4}$, we will obtain the next four-dimensional first-order differential system with 13 parameters:(1)$$\left\{\begin{array}{ccc} {\dot{x}}_{1} & = & a_{1}x_{1}-b_{1}x_{1}^{2}-c_{1}x_{1}x_{4}\\ {\dot{x}}_{2} & = & a_{2}x_{2}-b_{2}x_{2}^{2}-c_{2}x_{2}x_{4}\\ {\dot{x}}_{3} & = & a_{3}x_{3}-b_{3}x_{3}^{2}-c_{3}x_{3}x_{4}\\ {\dot{x}}_{4} & = & p_{4}x_{4}-p_{1}x_{1}x_{4}-p_{2}x_{2}x_{4}-p_{3}x_{3}x_{4} \end{array}\right.$$
with $a_{i}>0$, $b_{i}>0$ and $c_{i}>0$, $p_{i}>0$ for all i=1,2,3,4.

It is obvious that this model has medical relevance only when $x_{i}\geq 0$, i=1,2,3,4. Therefore, the solutions of the system lie in the set $\Sigma_{+}^{0}=\left\{\left(x_{1},x_{2},x_{2},x_{4}\right)\in R^{4}|x_{i}\geq 0\right\}$. Moreover, the hyperplanes, $\left\{x_{i}=0\right\}$, are invariant manifolds with respect to the flow of the system; i.e., any orbit starting from a point from $\Sigma_{+}=\left\{\left(x_{1},x_{2},x_{2},x_{4}\right)\in R^{4}|x_{i}>0\right\}$ remains in $\Sigma_{+}$. Then, the orbits cannot cross any of these four invariant hyperplanes. In conclusion, the study of this system in the zone with medical relevance $\Sigma_{+}$ is well-defined, which means that any orbit starting from the zone with medical relevance does not pass out of this zone, and any orbit starting from the zone without medical relevance does not enter into the zone with medical relevance.

## 3. The Local Analysis of the Mathematical Model

Next, we study the local behavior of the dynamical system defined by the first-order differential system (Equation 1). For this purpose, we will apply the classical theory of dynamical systems, namely the Hartman–Grobman theorem [19,20]. This famous theorem tells us that the behavior of a dynamical system in a neighborhood of a hyperbolic equilibrium point is qualitatively the same as the behavior of its linearized system near this equilibrium point. The linearized system of a dynamical system near an equilibrium point is exactly the linear system given by the associated Jacobi matrix at this equilibrium point. We recall that a hyperbolic equilibrium point means that no eigenvalue of the linearized system is equal to zero or has a real part equal to zero. Therefore, if we want to carry out a local study of such dynamical systems, then we can use the linearization of the system in order to analyze its behavior around equilibrium points.

Next, we summarize the methodology of the local analysis of a dynamical system using the following algorithm. If we consider a continuous dynamical system given by the nonlinear first-order differential system $\dot{x}=f\left(x\right)$, where $x=(x_{1},\ldots ,x_{n})$, $f\left(x\right)=\left(f_{1}\left(x\right),\ldots ,f_{n}\left(x\right)\right)$, and $f_{i}$ are smooth functions, we must follow the following steps:1.Determine the equilibrium points by solving the following system:
$$\left\{\begin{array}{c} f_{1}\left(x_{1},\ldots ,x_{n}\right)=0,\\ \cdots \cdots \cdots \cdots \cdots \\ f_{n}\left(x_{1},\ldots ,x_{n}\right)=0. \end{array}\right.$$In order to have biological meaning, we are interested in only finding equilibrium points $x=(x_{1},\ldots ,x_{n})$ with positive components, i.e., solutions with $x_{i}>0$, for all $i=\bar{1,n}$.2.At each equilibrium point $x=(x_{1},\ldots ,x_{n})$, we determine the Jacobian matrix $A={\left(\partial f_{i}/\partial x_{j}\right)}_{i,j=\bar{1,n}}\;$.3.At each equilibrium point $x=(x_{1},\ldots ,x_{n})$, we find the eigenvalues of Jacobian matrix *A*, which refers to the roots, $\lambda_{i}$, of the characteristic polynomial $\det\left(A-\lambda I_{n}\right)$.4.If all eigenvalues $\lambda_{i}$ of *A* are non-zero or have a non-zero real part, then the equilibrium point is hyperbolic, and the Hartman–Grobman theorem allows us to determine the local dynamics of the nonlinear system $\dot{x}=f\left(x\right)$ near this equilibrium point using the local behavior of linearized system $\dot{x}=Ax$ near the same equilibrium point. If not, the equilibrium is nonhyperbolic, and we must apply the center manifold theory to obtain some results about the local dynamics in the neighborhood of the nonhyperbolic equilibrium point.5.If all eigenvalues of *A* are negative or have a negative real part, then the equilibrium point is an attractor, which is also asymptotically stable (stable node or stable focus). Otherwise, the equilibrium point is unstable. More precisely, if all the eigenvalues of *A* are positive or have a positive real part, then the equilibrium point is a repeller (unstable node or unstable focus). If there are two eigenvalues with different signs, then the unstable equilibrium is called a saddle point.

The global analysis of the stability of the dynamical system can be carried out using a suitable Lyapunov function or other methods and techniques from the mathematical theory of dynamical systems [19,20]. This approach is not a part of the present study.

The Jacobi matrix at point $\left(x_{1},x_{2},x_{3},x_{4}\right)$ is
$$A=\left(\begin{array}{cccc} a_{1}-2b_{1}x_{1}-c_{1}x_{4} & 0 & 0 & -c_{1}x_{1}\\ 0 & a_{2}-2b_{2}x_{2}-c_{2}x_{4} & 0 & -c_{2}x_{2}\\ 0 & 0 & a_{3}-2b_{3}x_{3}-c_{3}x_{4} & -c_{3}x_{3}\\ -p_{1}x_{4} & -p_{2}x_{4} & -p_{3}x_{4} & p_{4}-p_{1}x_{1}-p_{2}x_{2}-p_{3}x_{3} \end{array}\right)$$

To find the equilibrium points of the system (Equation 1), we must solve the following system:(2)$$\left\{\begin{array}{c} x_{1}\left(a_{1}-b_{1}x_{1}-c_{1}x_{4}\right)=0\\ x_{2}\left(a_{2}-b_{2}x_{2}-c_{2}x_{4}\right)=0\\ x_{3}\left(a_{3}-b_{3}x_{3}-c_{3}x_{4}\right)=0\\ x_{4}\left(p_{4}-p_{1}x_{1}-p_{2}x_{2}-p_{3}x_{3}\right)=0 \end{array}\right.$$

Consequently, we obtain the following 15 equilibrium points:$$\begin{array}{cccc} E_{0}\left(0,0,0,0\right), & E_{1}^{1}\left(\frac{a_{1}}{b_{1}},0,0,0\right), & E_{1}^{2}\left(0,\frac{a_{2}}{b_{2}},0,0\right), & E_{1}^{3}\left(0,0,\frac{a_{3}}{b_{3}},0\right),\\ E_{2}^{12}\left(\frac{a_{1}}{b_{1}},\frac{a_{2}}{b_{2}},0,0\right), & E_{2}^{13}\left(\frac{a_{1}}{b_{1}},0,\frac{a_{3}}{b_{3}},0\right), & E_{2}^{23}\left(0,\frac{a_{2}}{b_{2}},\frac{a_{3}}{b_{3}},0\right), & E_{3}\left(\frac{a_{1}}{b_{1}},\frac{a_{2}}{b_{2}},\frac{a_{3}}{b_{3}},0\right), \end{array}$$
$$\begin{array}{ccc} E_{4}^{1}\left(\frac{p_{4}}{p_{1}},0,0,\frac{1}{c_{1}}\left(a_{1}-b_{1}\frac{p_{4}}{p_{1}}\right)\right), & E_{4}^{2}\left(0,\frac{p_{4}}{p_{2}},0,\frac{1}{c_{2}}\left(a_{2}-b_{2}\frac{p_{4}}{p_{2}}\right)\right), & E_{4}^{3}\left(0,0,\frac{p_{4}}{p_{3}},\frac{1}{c_{3}}\left(a_{3}-b_{3}\frac{p_{4}}{p_{3}}\right)\right), \end{array}$$
$$E_{5}^{12}\left(\frac{1}{b_{1}}\left(a_{1}-c_{1}v_{5}^{12}\right),\frac{1}{b_{2}}\left(a_{2}-c_{2}v_{5}^{12}\right),0,v_{5}^{12}\right)\;,v_{5}^{12}=\frac{1}{d_{5}^{12}}\left(p_{1}\frac{a_{1}}{b_{1}}+p_{2}\frac{a_{2}}{b_{2}}-p_{4}\right),d_{5}^{12}=p_{1}\frac{c_{1}}{b_{1}}+p_{2}\frac{c_{2}}{b_{2}},$$
$$E_{5}^{13}\left(\frac{1}{b_{1}}\left(a_{1}-c_{1}v_{5}^{13}\right),0,\frac{1}{b_{3}}\left(a_{3}-c_{3}v_{5}^{13}\right),v_{5}^{13}\right),v_{5}^{13}=\frac{1}{d_{5}^{13}}\left(p_{1}\frac{a_{1}}{b_{1}}+p_{3}\frac{a_{3}}{b_{3}}-p_{4}\right),d_{5}^{13}=p_{1}\frac{c_{1}}{b_{1}}+p_{3}\frac{c_{3}}{b_{3}},$$
$$E_{5}^{23}\left(0,\frac{1}{b_{2}}\left(a_{2}-c_{2}v_{5}^{23}\right),\frac{1}{b_{3}}\left(a_{3}-c_{3}v_{5}^{23}\right),v_{5}^{23}\right),v_{5}^{23}=\frac{1}{d_{5}^{23}}\left(p_{2}\frac{a_{2}}{b_{2}}+p_{3}\frac{a_{3}}{b_{3}}-p_{4}\right),d_{5}^{23}=p_{2}\frac{c_{2}}{b_{2}}+p_{3}\frac{c_{3}}{b_{3}}$$
and $E_{6}\left(x_{1},x_{2},x_{3},v_{6}\right)$, where $x_{1}=\frac{1}{b_{1}}\left(a_{1}-c_{1}v_{6}\right),x_{2}=\frac{1}{b_{2}}\left(a_{2}-c_{2}v_{6}\right),x_{3}=\frac{1}{b_{3}}\left(a_{3}-c_{3}v_{6}\right)$, and $v_{6}=\frac{1}{d_{6}}\left(p_{1}\frac{a_{1}}{b_{1}}+p_{2}\frac{a_{2}}{b_{2}}+p_{3}\frac{a_{3}}{b_{3}}-p_{4}\right)$, with $d_{6}=p_{1}\frac{c_{1}}{b_{1}}+p_{2}\frac{c_{2}}{b_{2}}+p_{3}\frac{c_{3}}{b_{3}}$.

Next, we will obtain the following results:

**Theorem** **1.**
*(i) $E_{0}$ is an unstable node;*

*(ii) $E_{1}^{1}$, $E_{1}^{2}$, and $E_{1}^{3}$ are saddle points whenever they belong to $\Sigma_{+}$;*

*(iii) $E_{2}^{12}$, $E_{2}^{13}$; $E_{2}^{23}$ are saddle points whenever they belong to $\Sigma_{+}$.*


**Proof.** (i) For $E_{0}\left(0,0,0,0\right)$, we have the Jacobian $A=\left(\begin{array}{cccc} a_{1} & 0 & 0 & 0\\ 0 & a_{2} & 0 & 0\\ 0 & 0 & a_{3} & 0\\ 0 & 0 & 0 & p_{4} \end{array}\right)$ with eigenvalues of $a_{1}$, $a_{2}$, $a_{3}$, $p_{4}>0$. Then, $E_{0}$ is a repeller; more exactly, it is an unstable node.(ii) For $E_{1}^{1}\left(\frac{a_{1}}{b_{1}},0,0,0\right)$, we have the Jacobian $A=\left(\begin{array}{cccc} -a_{1} & 0 & 0 & -c_{1}\frac{a_{1}}{b_{1}}\\ 0 & a_{2} & 0 & 0\\ 0 & 0 & a_{3} & 0\\ 0 & 0 & 0 & p_{4}-p_{1}\frac{a_{1}}{b_{1}} \end{array}\right)$ with eigenvalues of $-a_{1}$, $a_{2}$, $a_{3}$, $\frac{p_{4}b_{1}-p_{1}a_{1}}{b_{1}}=p_{4}-p_{1}\frac{a_{1}}{b_{1}}$; then, $E_{1}^{1}$ is a saddle point because all eigenvalues are real but have different signs.The same situation occurs for $E_{1}^{2}\left(0,\frac{a_{2}}{b_{2}},0,0\right)$, with the Jacobian $A=\left(\begin{array}{cccc} a_{1} & 0 & 0 & 0\\ 0 & -a_{2} & 0 & -c_{2}\frac{a_{2}}{b_{2}}\\ 0 & 0 & a_{3} & 0\\ 0 & 0 & 0 & p_{4}-p_{2}\frac{a_{2}}{b_{2}} \end{array}\right)$ and eigenvalues $a_{1}$, $-a_{2}$, $a_{3}$, $\frac{p_{4}b_{2}-p_{2}a_{2}}{b_{2}}=p_{4}-p_{2}\frac{a_{2}}{b_{2}}$, and for $E_{1}^{3}\left(0,0,\frac{a_{3}}{b_{3}},0\right)$, we have the Jacobian $A=\left(\begin{array}{cccc} a_{1} & 0 & 0 & 0\\ 0 & a_{2} & 0 & 0\\ 0 & 0 & -a_{3} & -c_{3}\frac{a_{3}}{b_{3}}\\ 0 & 0 & 0 & p_{4}-p_{3}\frac{a_{3}}{b_{3}} \end{array}\right)$, with eigenvalues $a_{1}$, $a_{2}$, $-a_{3}$, $\frac{p_{4}b_{3}-p_{3}a_{3}}{b_{3}}=p_{4}-p_{3}\frac{a_{3}}{b_{3}}$.(iii) For $E_{2}^{12}\left(\frac{a_{1}}{b_{1}},\frac{a_{2}}{b_{2}},0,0\right)$, we have the Jacobian $A=\left(\begin{array}{cccc} -a_{1} & 0 & 0 & -c_{1}\frac{a_{1}}{b_{1}}\\ 0 & -a_{2} & 0 & -c_{2}\frac{a_{2}}{b_{2}}\\ 0 & 0 & a_{3} & 0\\ 0 & 0 & 0 & p_{4}-p_{1}\frac{a_{1}}{b_{1}}-p_{2}\frac{a_{2}}{b_{2}} \end{array}\right)$, and eigenvalues $-a_{1}$, $-a_{2}$, $a_{3}$, $\frac{p_{4}b_{1}b_{2}-p_{1}a_{1}b_{2}-p_{2}a_{2}b_{1}}{b_{1}b_{2}}=p_{4}-p_{1}\frac{a_{1}}{b_{1}}-p_{2}\frac{a_{2}}{b_{2}}$. Then, $E_{2}^{12}$ is a saddle point. The same situation appears for $E_{2}^{13}\left(\frac{a_{1}}{b_{1}},0,\frac{a_{3}}{b_{3}},0\right)$ and $E_{2}^{23}\left(0,\frac{a_{2}}{b_{2}},\frac{a_{3}}{b_{3}},0\right)$.For $E_{2}^{13}$, we have $A=\left(\begin{array}{cccc} -a_{1} & 0 & 0 & -c_{1}\frac{a_{1}}{b_{1}}\\ 0 & a_{2} & 0 & 0\\ 0 & 0 & -a_{3} & -c_{3}\frac{a_{3}}{b_{3}}\\ 0 & 0 & 0 & p_{4}-p_{1}\frac{a_{1}}{b_{1}}-p_{3}\frac{a_{3}}{b_{3}} \end{array}\right)$ with eigenvalues $-a_{1}$, $a_{2}$, $-a_{3}$, $p_{4}-p_{1}\frac{a_{1}}{b_{1}}-p_{3}\frac{a_{3}}{b_{3}}$.For $E_{2}^{23}$, $A=\left(\begin{array}{cccc} a_{1} & 0 & 0 & 0\\ 0 & -a_{2} & 0 & -c_{2}\frac{a_{2}}{b_{2}}\\ 0 & 0 & -a_{3} & -c_{3}\frac{a_{3}}{b_{3}}\\ 0 & 0 & 0 & p_{4}-p_{2}\frac{a_{2}}{b_{2}}-p_{3}\frac{a_{3}}{b_{3}} \end{array}\right)$, and we have eigenvalues $a_{1}$, $-a_{2}$, $-a_{3}$, $p_{4}-p_{2}\frac{a_{2}}{b_{2}}-p_{3}\frac{a_{3}}{b_{3}}$. □

**Theorem** **2.**
*$E_{3}$ is an attractor (stable node) if and only if $p_{4}<p_{1}\frac{a_{1}}{b_{1}}+p_{2}\frac{a_{2}}{b_{2}}+p_{3}\frac{a_{3}}{b_{3}}$ and $E_{3}$ are a saddle point if and only if $p_{4}>p_{1}\frac{a_{1}}{b_{1}}+p_{2}\frac{a_{2}}{b_{2}}+p_{3}\frac{a_{3}}{b_{3}}$.*


**Proof.** At $E_{3}$, with $x_{1}=\frac{a_{1}}{b_{1}}$, $x_{2}=\frac{a_{2}}{b_{2}}$, $x_{3}=\frac{a_{3}}{b_{3}}$, $x_{4}=0$, we have the Jacobi matrix
$$A=\left(\begin{array}{cccc} -a_{1} & 0 & 0 & -c_{1}\frac{a_{1}}{b_{1}}\\ 0 & -a_{2} & 0 & -c_{2}\frac{a_{2}}{b_{2}}\\ 0 & 0 & -a_{3} & -c_{3}\frac{a_{3}}{b_{3}}\\ 0 & 0 & 0 & p_{4}-p_{1}\frac{a_{1}}{b_{1}}-p_{2}\frac{a_{2}}{b_{2}}-p_{3}\frac{a_{3}}{b_{3}} \end{array}\right)$$
with eigenvalues $\lambda_{1}=-a_{1}$, $\lambda_{2}=-a_{2}$, $\lambda_{3}=-a_{3}$, $\lambda_{4}=p_{4}-p_{1}\frac{a_{1}}{b_{1}}-p_{2}\frac{a_{2}}{b_{2}}-p_{3}\frac{a_{3}}{b_{3}}$. □

**Theorem** **3.**
*(i) $E_{4}^{1}$, $E_{4}^{2}$, and $E_{4}^{3}$ are saddle points whenever they belong to $\Sigma_{+}$;*

*(ii) $E_{5}^{12}$, $E_{5}^{13}$ and $E_{5}^{23}$ are saddle points whenever they belong to $\Sigma_{+}$.*


**Proof.** (i) For $E_{4}^{1}\left(\frac{p_{4}}{p_{1}},0,0,\frac{1}{c_{1}}\left(a_{1}-b_{1}\frac{p_{4}}{p_{1}}\right)\right)$, we have the Jacobian
$$A=\left(\begin{array}{cccc} -b_{1}\frac{p_{4}}{p_{1}} & 0 & 0 & -c_{1}\frac{p_{4}}{p_{1}}\\ 0 & a_{2}-\frac{c_{2}}{c_{1}}\left(a_{1}-b_{1}\frac{p_{4}}{p_{1}}\right) & 0\; & 0\\ 0 & 0 & a_{3}-\frac{c_{3}}{c_{1}}\left(a_{1}-b_{1}\frac{p_{4}}{p_{1}}\right)\; & 0\\ -\frac{p_{1}}{c_{1}}\left(a_{1}-b_{1}\frac{p_{4}}{p_{1}}\right) & -\frac{p_{2}}{c_{1}}\left(a_{1}-b_{1}\frac{p_{4}}{p_{1}}\right)\; & -\frac{p_{3}}{c_{1}}\left(a_{1}-b_{1}\frac{p_{4}}{p_{1}}\right) & 0 \end{array}\right)$$
with eigenvalues $a_{2}-\frac{c_{2}\left(p_{1}a_{1}-p_{4}b_{1}\right)}{c_{1}p_{1}}$, $a_{3}-\frac{c_{3}\left(p_{1}a_{1}-p_{4}b_{1}\right)}{c_{1}p_{1}}$, $\frac{1}{2p_{1}}\left(-p_{4}b_{1}+\sqrt{\Delta }\right)$, and $\frac{1}{2p_{1}}\left(-p_{4}b_{1}-\sqrt{\Delta }\right)$, where $\Delta =p_{4}^{2}b_{1}^{2}-4p_{1}p_{4}^{2}b_{1}+4p_{4}p_{1}^{2}a_{1}$. If $E_{4}^{1}$ is a proper equilibrium point (i.e., $\frac{1}{c_{1}}\left(a_{1}-b_{1}\frac{p_{4}}{p_{1}}\right)>0$), then $\Delta =p_{4}^{2}b_{1}^{2}+4p_{1}p_{4}\left(p_{1}a_{1}-p_{4}b_{1}\right)>0$ and $\sqrt{\Delta }>p_{4}b_{1}$. So, ithis results in $E_{4}^{1}$ being a saddle point because the last two eigenvalues have different signs.Following a similar analysis, $E_{4}^{2}\left(0,\frac{p_{4}}{p_{2}},0,\frac{1}{c_{2}}\left(a_{2}-b_{2}\frac{p_{4}}{p_{2}}\right)\right)$ and $E_{4}^{3}\left(0,0,\frac{p_{4}}{p_{3}},\frac{1}{c_{3}}\left(a_{3}-b_{3}\frac{p_{4}}{p3}\right)\right)$ are also saddle points whenever they belong to $\Sigma_{+}$.(ii) For $E_{5}^{12}\left(\frac{1}{b_{1}}\left(a_{1}-c_{1}v_{5}^{12}\right),\frac{1}{b_{2}}\left(a_{2}-c_{2}v_{5}^{12}\right),0,v_{5}^{12}\right)$, where $v_{5}^{12}=\frac{1}{d_{5}^{12}}\left(p_{1}\frac{a_{1}}{b_{1}}+p_{2}\frac{a_{2}}{b_{2}}-p_{4}\right)$ and $d_{5}^{12}=p_{1}\frac{c_{1}}{b_{1}}+p_{2}\frac{c_{2}}{b_{2}}$, we have the following Jacobian
$$A=\left(\begin{array}{cccc} -a_{1}+c_{1}v_{5}^{12} & 0 & 0 & -\frac{c_{1}}{b_{1}}\left(a_{1}-c_{1}v_{5}^{12}\right)\\ 0 & -a_{2}+c_{2}v_{5}^{12} & 0 & -\frac{c_{2}}{b_{2}}\left(a_{2}-c_{2}v_{5}^{12}\right)\\ 0 & 0 & a_{3}-c_{3}v_{5}^{12} & 0\\ -p_{1}v_{5}^{12} & -p_{2}v_{5}^{12} & -p_{3}v_{5}^{12} & 0 \end{array}\right)$$
or
$$A=\left(\begin{array}{cccc} -b_{1}x_{1} & 0 & 0 & -c_{1}x_{1}\\ 0 & -b_{2}x_{2} & 0 & -c_{2}x_{2}\\ 0 & 0 & a_{3}-c_{3}x_{4} & 0\\ -p_{1}x_{4} & -p_{2}x_{4} & -p_{3}x_{4} & 0 \end{array}\right)$$It is very complicated to find the eigenvalues of this matrix, but we observe the eigenvalue $\lambda_{1}=-\left(c_{3}x_{4}-a_{3}\right)$, where $x_{1}=\frac{a_{1}p_{2}c_{2}-c_{1}p_{2}a_{2}+c_{1}p_{4}b_{2}}{p_{1}c_{1}b_{2}+p_{2}c_{2}b_{1}}$, $x_{2}=\frac{a_{2}p_{1}c_{1}-c_{2}p_{1}a_{1}+c_{2}p_{4}b_{1}}{p_{1}c_{1}b_{2}+p_{2}c_{2}b_{1}}$, $x_{3}=0$, and $x_{4}=\frac{p_{1}a_{1}b_{2}+p_{2}a_{2}b_{1}-p_{4}b_{1}b_{2}}{p_{1}c_{1}b_{2}+p_{2}c_{2}b_{1}}$.The characteristic polynomial is $P_{E_{5}^{12}}\left(X\right)=X^{4}+A_{3}X^{3}+A_{2}X^{2}+A_{1}X+A_{0}$, where
$$\begin{array}{ccc} A_{3} & = & b_{1}x_{1}+b_{2}x_{2}+c_{3}x_{4}-a_{3}\\ A_{2} & = & b_{1}x_{1}c_{3}x_{4}+b_{2}x_{2}c_{3}x_{4}-p_{1}x_{4}c_{1}x_{1}-p_{2}x_{4}c_{2}x_{2}+b_{1}x_{1}b_{2}x_{2}-b_{2}x_{2}a_{3}-b_{1}x_{1}a_{3}\\ A_{1} & = & -b_{1}x_{1}b_{2}x_{2}a_{3}+b_{1}x_{1}b_{2}x_{2}c_{3}x_{4}-b_{1}x_{1}p_{2}x_{4}c_{2}x_{2}+p_{2}x_{4}c_{2}x_{2}a_{3}-p_{2}x_{4}^{2}c_{2}x_{2}c_{3}\\ \; & \; & -p_{1}x_{4}c_{1}x_{1}b_{2}x_{2}+p_{1}x_{4}c_{1}x_{1}a_{3}-p_{1}x_{4}^{2}c_{1}x_{1}c_{3}\\ A_{0} & = & x_{2}x_{1}x_{4}\left(a_{3}-c_{3}x_{4}\right)\left(b_{1}p_{2}c_{2}+p_{1}c_{1}b_{2}\right) \end{array}$$According to the Routh–Hurwitz criterion for fourth-order polynomials, the polynomial
$$P\left(X\right)=X^{4}+A_{3}X^{3}+A_{2}X^{2}+A_{1}X+A_{0}$$
has all roots in the open left half-plane (i.e., $\lambda_{i}<0$ or Re $\lambda_{i}<0$, for all *i*) if and only if
$$A_{3}>0,\;\;A_{2}A_{3}-A_{1}>0,\;\;A_{1}A_{2}A_{3}-A_{0}A_{3}^{2}-A_{1}^{2}>0\;\;\mathrm{and}\;\;A_{0}>0.$$We are interested in studying equilibrium $E_{5}^{12}\left(x_{1},x_{2},x_{3},x_{4}\right)$ only if it has components that are all positive, which means $x_{4}<\frac{a_{1}}{c_{1}}$, $x_{4}<\frac{a_{2}}{c_{2}}$, $p_{4}<p_{1}\frac{a_{1}}{b_{1}}+p_{2}\frac{a_{2}}{b_{2}}$; then, we obtain $A_{0}<0$ if and only if $x_{4}>\frac{a_{3}}{c_{3}}$ or $A_{0}>0$ if and only if $x_{4}<\frac{a_{3}}{c_{3}}$. If $\lambda_{i}$ (i=1,2,3,4) denotes the eigenvalues at $E_{5}^{12}$, then, according to Viéte’s relations, we have $\lambda_{1}\lambda_{2}\lambda_{3}\lambda_{4}=A_{0}$. Thus, it can be said that $E_{5}^{12}$ is a saddle point if $x_{4}>\frac{a_{3}}{c_{3}}$.If $x_{4}=\frac{a_{3}}{c_{3}}$ (i.e., $A_{0}=0$), then $x_{1}=\frac{1}{b_{1}}\left(a_{1}-c_{1}\frac{a_{3}}{c_{3}}\right)$ and $x_{2}=\frac{1}{b_{2}}\left(a_{2}-c_{2}\frac{a_{3}}{c_{3}}\right)$, and this results in the following:
$$\begin{array}{ccc} A_{3} & = & b_{1}x_{1}+b_{2}x_{2}+c_{3}x_{4}-a_{3}=\left(a_{1}-c_{1}\frac{a_{3}}{c_{3}}\right)+\left(a_{2}-c_{2}\frac{a_{3}}{c_{3}}\right)>0,\\ A_{2} & = & b_{1}x_{1}c_{3}x_{4}+b_{2}x_{2}c_{3}x_{4}-p_{1}x_{4}c_{1}x_{1}-p_{2}x_{4}c_{2}x_{2}+b_{1}x_{1}b_{2}x_{2}-b_{2}x_{2}a_{3}-b_{1}x_{1}a_{3}\\ & = & -p_{1}\frac{a_{3}}{c_{3}}\frac{c_{1}}{b_{1}}\left(a_{1}-c_{1}\frac{a_{3}}{c_{3}}\right)-p_{2}\frac{a_{3}}{c_{3}}\frac{c_{2}}{b_{2}}\left(a_{2}-c_{2}\frac{a_{3}}{c_{3}}\right)+\left(a_{1}-c_{1}\frac{a_{3}}{c_{3}}\right)\left(a_{2}-c_{2}\frac{a_{3}}{c_{3}}\right),\\ A_{1} & = & -b_{1}x_{1}b_{2}x_{2}a_{3}+b_{1}x_{1}b_{2}x_{2}c_{3}x_{4}-b_{1}x_{1}p_{2}x_{4}c_{2}x_{2}+p_{2}x_{4}c_{2}x_{2}a_{3}\\ & - & p_{2}x_{4}^{2}c_{2}x_{2}c_{3}-p_{1}x_{4}c_{1}x_{1}b_{2}x_{2}+p_{1}x_{4}c_{1}x_{1}a_{3}-p_{1}x_{4}^{2}c_{1}x_{1}c_{3}\\ & = & -\left(a_{1}-c_{1}\frac{a_{3}}{c_{3}}\right)p_{2}\frac{a_{3}}{c_{3}}\frac{c_{2}}{b_{2}}\left(a_{2}-c_{2}\frac{a_{3}}{c_{3}}\right)-p_{1}\frac{a_{3}}{c_{3}}\frac{c_{1}}{b_{1}}\left(a_{1}-c_{1}\frac{a_{3}}{c_{3}}\right)\left(a_{2}-c_{2}\frac{a_{3}}{c_{3}}\right)<0, \end{array}$$Since the eigenvalues of the characteristic polynomial at the nonhyperbolic equilibrium point $E_{5}^{12}$ are $\lambda_{1}=0$, $\lambda_{2}$, $\lambda_{3}$, and $\lambda_{4}$, where the last three are exactly the roots of the polynomial $Q\left(X\right)=X^{3}+A_{3}X^{2}+A_{2}X^{1}+A_{1}$, the following
$$\lambda_{2}+\lambda_{3}+\lambda_{4}=-A_{3}<0\;,\lambda_{2}\lambda_{3}+\lambda_{3}\lambda_{4}+\lambda_{4}\lambda_{2}=A_{2}\;,\lambda_{2}\lambda_{3}\lambda_{4}=-A_{1}>0\;\mathrm{and}\;$$
$$\lambda_{2}^{2}+\lambda_{3}^{2}+\lambda_{4}^{2}={\left(\lambda_{2}+\lambda_{3}+\lambda_{4}\right)}^{2}-2\left(\lambda_{2}\lambda_{3}+\lambda_{3}\lambda_{4}+\lambda_{4}\lambda_{2}\right)=A_{3}^{2}-2A_{2}=$$
$${\left(a_{1}-c_{1}\frac{a_{3}}{c_{3}}+a_{2}-c_{2}\frac{a_{3}}{c_{3}}\right)}^{2}-2\left[-p_{1}\frac{a_{3}}{c_{3}}\frac{c_{1}}{b_{1}}\left(a_{1}-c_{1}\frac{a_{3}}{c_{3}}\right)-p_{2}\frac{a_{3}}{c_{3}}\frac{c_{2}}{b_{2}}\left(a_{2}-c_{2}\frac{a_{3}}{c_{3}}\right)+\right.$$
$$\left.\left(a_{1}-c_{1}\frac{a_{3}}{c_{3}}\right)\left(a_{2}-c_{2}\frac{a_{3}}{c_{3}}\right)\right]={\left(a_{1}-c_{1}\frac{a_{3}}{c_{3}}\right)}^{2}+{\left(a_{2}-c_{2}\frac{a_{3}}{c_{3}}\right)}^{2}+2p_{1}\frac{a_{3}}{c_{3}}\frac{c_{1}}{b_{1}}\left(a_{1}-c_{1}\frac{a_{3}}{c_{3}}\right)+$$
$$2p_{2}\frac{a_{3}}{c_{3}}\frac{c_{2}}{b_{2}}\left(a_{2}-c_{2}\frac{a_{3}}{c_{3}}\right)>0\;\mathrm{occurs}\;\mathrm{due}\;\mathrm{to}\;\mathrm{the}\;\mathrm{positivity}\;\mathrm{of}\;\mathrm{the}\;\mathrm{first}\;\mathrm{two}\;\mathrm{components}\;\mathrm{of}\;E_{5}^{12}.$$Then, $\lambda_{2}$, $\lambda_{3}$, and $\lambda_{4}$ are real eigenvalues with different signs (two negative and one positive), which means that $E_{5}^{12}$ is also a saddle point in this case.If $x_{4}<\frac{a_{3}}{c_{3}}$, then $A_{0}>0$, but the sign of $A_{1}$, $A_{2}$, and $A_{3}$ is very complicated to study because we have too many parameters. However, by using the first root $\lambda_{1}=-\left(c_{3}x_{4}-a_{3}\right)$ of the characteristic polynomial at $E_{5}^{12}$, we have
$$P_{E_{5}^{12}}\left(X\right)=X^{4}+A_{3}X^{3}+A_{2}X^{2}+A_{1}X+A_{0}=\left(X-\lambda_{1}\right)\left(X^{3}+B_{2}X^{2}+B_{1}X+B_{0}\right),$$
where $B_{2}=A_{3}+\lambda_{1}$, $B_{1}=A_{2}+\lambda_{1}B_{2}$, $B_{0}=A_{1}+\lambda_{1}B_{1}$, or $A_{0}=-\lambda_{1}B_{0}$. However, in this case, we have $x_{4}<\frac{a_{3}}{c_{3}}$; then, $\lambda_{1}=-\left(c_{3}x_{4}-a_{3}\right)>0$ results and then $B_{0}<0$ because $A_{0}>0$. It follows that $\lambda_{2}\lambda_{3}\lambda_{4}=B_{0}<0$, which means that $E_{5}^{12}$ is also a saddle point in this case.Then, $E_{5}^{12}$ is a saddle point whenever it belongs to $\Sigma_{+}$.Similarly, we have the same results for $E_{5}^{13}\left(\frac{1}{b_{1}}\left(a_{1}-c_{1}v_{5}^{13}\right),0,\frac{1}{b_{3}}\left(a_{3}-c_{3}v_{5}^{13}\right),v_{5}^{13}\right)$, where
$v_{5}^{13}=\frac{1}{d_{5}^{13}}\left(p_{1}\frac{a_{1}}{b_{1}}+p_{3}\frac{a_{3}}{b_{3}}-p_{4}\right)$, $d_{5}^{13}=p_{1}\frac{c_{1}}{b_{1}}+p_{3}\frac{c_{3}}{b_{3}}$. For $E_{5}^{23}(0,\frac{1}{b_{2}}\left(a_{2}-c_{2}v_{5}^{23}\right),\frac{1}{b_{3}}\left(a_{3}-c_{3}v_{5}^{23}\right),$
$v_{5}^{23})$, where $v_{5}^{23}=\frac{1}{d_{5}^{23}}\left(p_{2}\frac{a_{2}}{b_{2}}+p_{3}\frac{a_{3}}{b_{3}}-p_{4}\right)$, $d_{5}^{23}=p_{2}\frac{c_{2}}{b_{2}}+p_{3}\frac{c_{3}}{b_{3}}$, we have the same results. Then, these two equilibrium points are saddle points. □

Finally, for $E_{6}\left(x_{1},x_{2},x_{3},x_{4}\right)$, where $x_{1}=$ $\frac{1}{b_{1}}\left(a_{1}-c_{1}v_{6}\right)$, $x_{2}=$ $\frac{1}{b_{2}}\left(a_{2}-c_{2}v_{6}\right)$, $x_{3}=\frac{1}{b_{3}}\left(a_{3}-c_{3}v_{6}\right)$, and $x_{4}=v_{6}=\frac{1}{d_{6}}\left(p_{1}\frac{a_{1}}{b_{1}}+p_{2}\frac{a_{2}}{b_{2}}+p_{3}\frac{a_{3}}{b_{3}}-p_{4}\right)$, with $d_{6}=$
$p_{1}\frac{c_{1}}{b_{1}}+p_{2}\frac{c_{2}}{b_{2}}+$
$p_{3}\frac{c_{3}}{b_{3}}$, we obtain the Jacobian
$$A=\left(\begin{array}{cccc} -b_{1}x_{1} & 0 & 0 & -c_{1}x_{1}\\ 0 & -b_{2}x_{2} & 0 & -c_{2}x_{2}\\ 0 & 0 & -b_{3}x_{3} & -c_{3}x_{3}\\ -p_{1}x_{4} & -p_{2}x_{4} & -p_{3}x_{4} & 0 \end{array}\right)$$
or
$$A=\left(\begin{array}{cccc} -a_{1}+\frac{c_{1}}{d_{6}}t_{6} & 0 & 0 & -\frac{c_{1}}{b_{1}}\left(a_{1}-\frac{c_{1}}{d_{6}}t_{6}\right)\\ 0 & -a_{2}+\frac{c_{2}}{d_{6}}t_{6} & 0 & -\frac{c_{2}}{b_{2}}\left(a_{2}-\frac{c_{2}}{d_{6}}t_{6}\right)\\ 0 & 0 & -a_{3}+\frac{c_{3}}{d_{6}}t_{6} & -\frac{c_{3}}{b_{3}}\left(a_{3}-\frac{c_{3}}{d_{6}}t_{6}\right)\\ -\frac{p_{1}}{d_{6}}t_{6} & -\frac{p_{2}}{d_{6}}t_{6} & -\frac{p_{3}}{d_{6}}t_{6} & 0 \end{array}\right)\;,$$
where $t_{6}=p_{1}\frac{a_{1}}{b_{1}}+p_{2}\frac{a_{2}}{b_{2}}+p_{3}\frac{a_{3}}{b_{3}}-p_{4}$.

The characteristic polynomial is $P_{E_{6}}\left(X\right)=X^{4}+A_{3}X^{3}+A_{2}X^{2}+A_{1}X+A_{0}$, where
$$\begin{array}{ccc} A_{3} & = & b_{1}x_{1}+b_{2}x_{2}+b_{3}x_{3}\\ A_{2} & = & b_{1}x_{1}b_{2}x_{2}+b_{1}x_{1}b_{3}x_{3}+b_{2}x_{2}b_{3}x_{3}-\left(p_{1}c_{1}x_{1}+p_{2}c_{2}x_{2}+c_{3}x_{3}p_{3}\right)x_{4}\\ A_{1} & = & x_{1}x_{2}x_{3}b_{1}b_{2}b_{3}-x_{2}x_{3}x_{4}\left(p_{2}b_{3}c_{2}+p_{3}b_{2}c_{3}\right)-x_{1}x_{2}x_{4}\left(p_{1}b_{2}c_{1}+p_{2}b_{1}c_{2}\right)\\ \; & - & x_{1}x_{3}x_{4}\left(p_{3}b_{1}c_{3}+p_{1}b_{3}c_{1}\right)\\ A_{0} & = & -x_{1}x_{2}x_{3}x_{4}\left(p_{3}b_{1}b_{2}c_{3}+p_{1}b_{2}b_{3}c_{1}+b_{1}p_{2}b_{3}c_{2}\right) \end{array}$$

Due to the fact that we are interested in only studying the proper equilibrium points, all components of $E_{6}\left(x_{1},x_{2},x_{3},x_{4}\right)$ are positive, which means $a_{1}-c_{1}x_{4}>0$, $a_{2}-c_{2}x_{4}>0$, $a_{3}-c_{3}x_{4}>0$, and $p_{1}\frac{a_{1}}{b_{1}}+p_{2}\frac{a_{2}}{b_{2}}+p_{3}\frac{a_{3}}{b_{3}}-p_{4}>0$; thus, we have $A_{3}>0$ and $A_{0}<0$.

Then, we obtain the next main theorem:

**Theorem** **4.**
*(i) The equilibrium point $E_{6}$ is a saddle whenever it belongs to $\Sigma_{+}$.*

*(ii) The system (Equation 1) does not undergo a Hopf–Hopf bifurcation at $E_{6}$ on $\Sigma_{+}$.*

*(iii) The equilibrium point $E_{6}$ bifurcates from $E_{3}$ along the hyperplane via a transcritical bifurcation:*

$$S=\{\left(p_{1},p_{2},p_{3},p_{4}\right)|p_{1}\frac{a_{1}}{b_{1}}+p_{2}\frac{a_{2}}{b_{2}}+p_{3}\frac{a_{3}}{b_{3}}-p_{4}=0\}$$


*(iv) Three more transcritical bifurcations arise in the system (Equation 1) on the hyperplanes:*

$$S_{12}=\left\{\left(p_{1},p_{2},p_{3},p_{4}\right)|p_{1}\left(\frac{a_{1}}{b_{1}}-\frac{a_{3}c_{1}}{b_{1}c_{3}}\right)+p_{2}\left(\frac{a_{2}}{b_{2}}-\frac{a_{3}c_{2}}{b_{2}c_{3}}\right)-p_{4}=0;\begin{array}{c} a_{1}c_{3}-a_{3}c_{1}>0\\ a_{2}c_{3}-a_{3}c_{2}>0 \end{array}\right\}\;,$$


$$S_{13}=\left\{\left(p_{1},p_{2},p_{3},p_{4}\right)|p_{1}\left(\frac{a_{1}}{b_{1}}-\frac{a_{2}c_{1}}{b_{1}c_{2}}\right)+p_{3}\left(\frac{a_{3}}{b_{3}}-\frac{a_{2}c_{3}}{b_{3}c_{2}}\right)-p_{4}=0;\begin{array}{c} a_{1}c_{2}-a_{2}c_{1}>0\\ a_{3}c_{2}-a_{2}c_{3}>0 \end{array}\right\}\;,$$


$$S_{23}=\left\{\left(p_{1},p_{2},p_{3},p_{4}\right)|p_{2}\left(\frac{a_{2}}{b_{2}}-\frac{a_{1}c_{2}}{b_{2}c_{1}}\right)+p_{3}\left(\frac{a_{3}}{b_{3}}-\frac{a_{1}c_{3}}{b_{3}c_{1}}\right)-p_{4}=0;\begin{array}{c} a_{2}c_{1}-a_{1}c_{2}>0\\ a_{3}c_{1}-a_{1}c_{3}>0 \end{array}\right\}\;.$$

*where $E_{6}$ collides with $E_{5}^{12}$ on $S_{12}$, $E_{6}$ collides with $E_{5}^{13}$ on $S_{13}$, and $E_{6}$ collides with $E_{5}^{23}$ on $S_{23}$, respectively.*

*(v) Moreover, there are three transcritical bifurcations that arise in the system (Equation 1) on the two planes at $\pi_{1}=S_{12}\cap S_{13}$, $\pi_{2}=S_{12}\cap S_{23}$, and $\pi_{3}=S_{13}\cap S_{23}$. More precisely, $E_{6}$ collides with $E_{4}^{1}$ on $\pi_{1}$, $E_{6}$ collides with $E_{4}^{2}$ on $\pi_{2}$, and $E_{6}$ collides with $E_{4}^{3}$ on $\pi_{3}$, respectively.*


**Proof.** (i) If we suppose that $E_{6}$ is an attractor, then all roots of the characteristic polynomial at $E_{6}$ are in the open left half-plane (i.e., $\lambda_{i}<0$ or Re $\lambda_{i}<0$, for all *i*); then, according to the Routh–Hurwitz criterion, we must have $A_{3}>0$, $A_{2}A_{3}-A_{1}>0$, $A_{1}A_{2}A_{3}-A_{0}A_{3}^{2}-A_{1}^{2}>0$, and $A_{0}>0$. However, $A_{0}<0$ and $E_{6}$ cannot be attractors. On the other hand, if we suppose that $E_{6}$ is a repeller, with $\lambda_{i}>0$ for all *i*, we obtain a contradiction again with respect to $\lambda_{1}\lambda_{2}\lambda_{3}\lambda_{4}=A_{0}<0$, according with the relations of Viéte. In conclusion, $E_{6}$ is not a repeller, and we can conclude that $E_{6}$ is a saddle point whenever it belongs to $\Sigma_{+}$.(ii) If we suppose that the characteristic polynomial at $E_{6}$ has roots $\pm i\omega_{1}$ and $\pm i\omega_{2}$, with $\omega_{1}>0$ and $\omega_{2}>0$, then $\lambda_{1}+\lambda_{2}+\lambda_{3}+\lambda_{4}=0$ and $\lambda_{1}\lambda_{2}\lambda_{3}\lambda_{4}=\omega_{1}^{2}\omega_{2}^{2}>0$, which contradicts the relations of Viéte ($A_{3}>0$ and $A_{0}<0$). Therefore, system (Equation 1) does not undergo a Hopf–Hopf bifurcation at $E_{6}$ in $\Sigma_{+}$.(iii) It is obvious that $E_{6}$ coincides with $E_{3}$ on hyperplane *S*. In order to prove that a transcritical bifurcation takes place on *S*, we will apply Sotomayor’s theorem [20].For this purpose, we will assume that $a_{i}/b_{i}$, $p_{i}$, and i=1,2,3 are constants while $p_{4}$ can vary. Let us use parameter $\mu =p_{4}-k$, where $k=\sum_{i=1}^{3}pi\frac{a_{i}}{b_{i}}$, and let us denote $u={\left(x_{1},x_{2},x_{3},x_{4}\right)}^{T}$, $F={\left(F_{1},F_{2},F_{3},F_{4}\right)}^{T}$, where *T* denotes the transpose matrix, i.e., ${\left(x_{1},x_{2},x_{3},x_{4}\right)}^{T}=\left(\begin{array}{c} x_{1}\\ x_{2}\\ x_{3}\\ x_{4} \end{array}\right)$. Then, the system (Equation 1) can be written in the following form:
(3)$$\dot{u}=F\left(u,\mu \right),$$
where $F_{1}=a_{1}x_{1}-b_{1}x_{1}^{2}-c_{1}x_{1}x_{4}$, $F_{2}=a_{2}x_{2}-b_{2}x_{2}^{2}-c_{2}x_{2}x_{4}$, $F_{3}=a_{3}x_{3}-b_{3}x_{3}^{2}-c_{3}x_{3}x_{4}$, and $F_{4}=\left(\mu +k\right)x_{4}-p_{1}x_{1}x_{4}-p_{2}x_{2}x_{4}-p_{3}x_{3}x_{4}$.Next, we denote $F_{\mu }={\left(\frac{\partial F_{1}}{\partial \mu },\frac{\partial F_{2}}{\partial \mu },\frac{\partial F_{3}}{\partial \mu },\frac{\partial F_{4}}{\partial \mu }\right)}^{T}$ and DF(u,μ) as the Jacobian matrix of *F* from (Equation 3) with respect to *u*. If we set $u_{0}=E_{3}=\left(\frac{a_{1}}{b_{1}},\frac{a_{2}}{b_{2}},\frac{a_{3}}{b_{3}},0\right)$, and $\mu_{0}=0$, then 0 is an eigenvalue both for $DF(u_{0},\mu_{0})$ and transpose $DF^{T}\left(u,\mu \right)$, with the corresponding eigenvectors $v={\left(-\frac{c_{1}}{b_{1}},-\frac{c_{2}}{b_{2}},-\frac{c_{3}}{b_{3}},1\right)}^{T}$ for $DF(u_{0},\mu_{0})$ and $w={\left(0,0,0,1\right)}^{T}$ for $DF^{T}\left(u_{0},\mu_{0}\right)$.The first condition of the transcritical bifurcation, $w^{T}F_{\mu }\left(u_{0},\mu_{0}\right)=0$, is obviously satisfied.If we denote the Jacobi matrix of $F_{\mu }={\left(0,0,0,x_{4}\right)}^{T}$ by $DF_{\mu }$, then we fulfill the condition $w^{T}\left[DF_{\mu }\left(u_{0},\mu_{0}\right)v\right]=1\neq 0$.In order to prove the third condition, we denote the second differential of *F* at (u,u) by
$$D^{2}F\left(u,u\right)={\left(d^{2}F_{1}\left(u,u\right)\;,d^{2}F_{2}\left(u,u\right)\;,d^{2}F_{3}\left(u,u\right)\;,d^{2}F_{4}\left(u,u\right)\right)}^{T}\;,$$
where $d^{2}F_{i}\left(u,u\right)$ is the second-order differential of $F_{i}$ applied at the pair (u,u), $u={\left(x_{1}\;,x_{2}\;,x_{3}\;,x_{4}\right)}^{T}$. Then, $d^{2}F_{i}\left(u,u\right)=-2b_{i}x_{i}^{2}-2c_{i}x_{i}x_{4}$, for i=1,2,3, $d^{2}F_{4}\left(u,u\right)=-2\left(p_{1}x_{1}+p_{2}x_{2}+p_{3}x_{3}\right)x_{4}$; then, at $\left(u_{0},\mu_{0}\right)$, we have
$$w^{T}\left[D^{2}F\left(u_{0},\mu_{0}\right)\left(v,v\right)\right]=2\left(p_{1}\frac{c_{1}}{b_{1}}+p_{2}\frac{c_{2}}{b_{2}}+p_{3}\frac{c_{3}}{b_{3}}\right)\neq 0\;,$$
which means that we have checked the last condition for a transcritical bifurcation on *S*:(iv) We have similar treatments to those stated above for each hyperplane.(v) By using varying parameters for $\mu_{1}$ and $\mu_{2}$ for each of the two planes, we can check the conditions of the Sotomayor theorem. □

## 4. Theoretical Medical Interpretations of the Results

By taking into account that this four-dimensional mathematical model has 15 equilibrium points, one of which is a repeller (unstable node), another is an attractor or saddle, and the rest of the 13 are saddle points, the aim of this section is to present some theoretical medical interpretations of the behavior of the system near each equilibrium point. Similar interpretations were provided in [6] for a three-dimensional model.

**Interpretation for equilibrium point** $E_{0}\left(0,0,0,0\right)$.

Because $E_{0}$ is a repeller (unstable node) with strictly positive eigenvalues—$a_{1}$, $a_{2}$, $a_{3}$, and $p_{4}$—it follows that any orbit $x\left(t\right)=(x_{1}\left(t\right),x_{2}\left(t\right),x_{3}\left(t\right),x_{4}\left(t\right))$ starting at a point $x_{0}=\left(x_{10},x_{20},x_{30},x_{40}\right)\in \Sigma_{+}$ close to $E_{0}$ will depart from it when *t* is large; that is, $x_{4}\left(t\right)$ may escape to infinity. Moreover, since the Hopf bifurcation is not possible at $E_{0}$ (its eigenvalues are real), a stable limit cycle surrounding $E_{0}$ does not arise by such a bifurcation. When taken together, these results mean that if the human immune system is sufficiently weak when the virus starts to proliferate in the body, then the virus can win the fight using antibody cells; from a theoretical point of view, this means the death of the human body. We emphasize the idea that viral cells can increase (via uncontrolled growth) even if they were present in a negligible amount at first. However, the levels of innate immunity, humoral immunity, and cellular immunity are extremely small within the vicinity of the origin.

**Interpretation for** $E_{1}^{1}\left(\frac{a_{1}}{b_{1}},0,0,0\right)$, $E_{1}^{2}\left(0,\frac{a_{2}}{b_{2}},0,0\right)$ and $E_{1}^{3}\left(0,0,\frac{a_{3}}{b_{3}},0\right)$.

If we consider equilibrium point $E_{1}^{1}\left(\frac{a_{1}}{b_{1}},0,0,0\right)$, then its eigenvalues are $-a_{1}$, $a_{2}$, $a_{3}$, and $p_{4}-p_{1}\frac{a_{1}}{b_{1}}$, and $E_{1}^{1}$ is a saddle point for either $p_{4}-p_{1}\frac{a_{1}}{b_{1}}>0$ or $p_{4}-p_{1}\frac{a_{1}}{b_{1}}<0$. Any orbit x(t) starting at point $x_{0}\in \Sigma_{+}$ close to $E_{1}^{1}$, $x_{0}\notin W_{E_{1}^{1}}^{s}$, will depart from $E_{1}^{1}$ when *t* is large; that is, $x_{4}\left(t\right)$ may escape to infinity. A stable limit cycle around $E_{1}^{1}$ cannot arise via a Hopf bifurcation since all eigenvalues are real. This means that if humoral immunity and cellular immunity are not at the normal level when the virus invades the human body, then the virus may win even though the level of innate immunity is normal, even at a maximum threshold value of $a_{1}/b_{1}$. Thus, a deficiency in the two immunity types may result in the virus’s victory. Viral cells can increase even if they are present in a negligible amount at first. For $E_{1}^{2}$ and $E_{1}^{3}$, the results are similar.

**Interpretation for** $E_{2}^{12}\left(\frac{a_{1}}{b_{1}},\frac{a_{2}}{b_{2}},0,0\right)$, $E_{2}^{13}\left(\frac{a_{1}}{b_{1}},0,\frac{a_{3}}{b_{3}},0\right)$, and $E_{2}^{23}\left(0,\frac{a_{2}}{b_{2}},\frac{a_{3}}{b_{3}},0\right)$.

If we consider equilibrium point $E_{2}^{12}\left(\frac{a_{1}}{b_{1}},\frac{a_{2}}{b_{2}},0,0\right)$, then its eigenvalues are $-a_{1}$, $-a_{2}$, $a_{3}$, and $p_{4}-p_{1}\frac{a_{1}}{b_{1}}-p_{2}\frac{a_{2}}{b_{2}}$, and $E_{1}^{1}$ is a saddle point for either $p_{4}-p_{1}\frac{a_{1}}{b_{1}}-p_{2}\frac{a_{2}}{b_{2}}>0$ or $p_{4}-p_{1}\frac{a_{1}}{b_{1}}-p_{2}\frac{a_{2}}{b_{2}}<0$. Any orbit x(t) starting at a point $x_{0}\in \Sigma_{+}$ close to $E_{1}^{12}$, $x_{0}\notin W_{E_{1}^{12}}^{s}$ will depart from $E_{1}^{12}$ when *t* is large; that is, $x_{4}\left(t\right)$ may escape to infinity. A stable limit cycle around $E_{1}^{12}$ cannot arise via a Hopf bifurcation since all eigenvalues are real. This means that if cellular immunity is not at the normal level when the virus invades the human body, then the virus may win even when innate immunity and humoral immunity are at normal levels and even at the maximum threshold values of $a_{1}/b_{1}$ and $a_{2}/b_{2}$, respectively. Thus, a deficiency in the quantity of a single type of immunity may lead to the virus’s victory. Viral cells can increase even if they were present in a negligible amount at first. For $E_{2}^{13}$ and $E_{2}^{23}$, the results are similar.

**Interpretation for** $E_{3}\left(\frac{a_{1}}{b_{1}},\frac{a_{2}}{b_{2}},\frac{a_{3}}{b_{3}},0\right)$.

If $p_{4}<p_{1}\frac{a_{1}}{b_{1}}+p_{2}\frac{a_{2}}{b_{2}}+p_{3}\frac{a_{3}}{b_{3}}$, then $E_{3}$ is an attractor (stable node). It follows that any orbit x(t) starting at a point $x_{0}\in \Sigma_{+}$ close to $E_{3}$ will converge to $\left(\frac{a_{1}}{b_{1}},\frac{a_{2}}{b_{2}},\frac{a_{3}}{b_{3}},0\right)$ when *t* is large, which means that $x_{4}\left(t\right)$ tends toward 0 when *t* tends toward +∞. In conclusion, this model shows us that if innate immunity, humoral immunity, and cellular immunity are at normal levels (i.e., $a_{i}/b_{i}$, i=1,2,3) during the first moment the virus is discovered and if their joint actions kill the virus at a higher rate than the rate of viral proliferation (i.e., $p_{1}\frac{a_{1}}{b_{1}}+p_{2}\frac{a_{2}}{b_{2}}+p_{3}\frac{a_{3}}{b_{3}}>p_{4}$), then the immune system of the body has the ability to stop the virus’s multiplication and can liquidate it; then, the immune system wins the fight. Otherwise, if the joint destruction rate $p_{1}\frac{a_{1}}{b_{1}}+p_{2}\frac{a_{2}}{b_{2}}+p_{3}\frac{a_{3}}{b_{3}}$ of the virus exhibited by the three immunity system’s actions is not sufficiently strong for overcoming the rate $p_{4}$ of the virus’s proliferation ($p_{1}\frac{a_{1}}{b_{1}}+p_{2}\frac{a_{2}}{b_{2}}+p_{3}\frac{a_{3}}{b_{3}}<p_{4}$, i.e., $E_{3}$ is a saddle point), then the virus may win because $x_{4}\left(t\right)$ may escape to infinity if an orbit x(t) starts at a point $x_{0}\in \Sigma_{+}$ close to $E_{3}$, but $x_{0}\notin W_{E_{3}}^{s}$. Otherwise, if $x_{0}\in W_{E_{3}}^{s}$, then the quantity of virus $x_{4}\left(t\right)$ will converge to 0 when *t* is large, and the virus is eliminated. A Hopf bifurcation that leads to a stable cycle around $E_{3}$ is not possible because all eigenvalues are real.

**Interpretation for** $E_{4}^{1}$, $E_{4}^{2}$, and $E_{4}^{3}$.

If $p_{1}\frac{a_{1}}{b_{1}}>p_{4}$, then $E_{4}^{1}\left(\frac{p_{4}}{p_{1}},0,0,\frac{1}{c_{1}}\left(a_{1}-b_{1}\frac{p_{4}}{p_{1}}\right)\right)\in \Sigma_{+}$, and at this equilibrium point, no Hopf bifurcation is possible because all eigenvalues are real. Due to their different signs, $E_{4}^{1}$ is a saddle, and $x_{4}\left(t\right)$ may escape to infinity, which means that the virus may win. For $E_{4}^{2}$ and $E_{4}^{3}$, we have the same scenario.

**Interpretation for** $E_{5}^{12}$, $E_{5}^{13}$, and $E_{5}^{23}$.

For $E_{5}^{12}\left(\frac{1}{b_{1}}\left(a_{1}-c_{1}v_{5}^{12}\right),\frac{1}{b_{2}}\left(a_{2}-c_{2}v_{5}^{12}\right),0,v_{5}^{12}\right)$, where $v_{5}^{12}=\frac{1}{d_{5}^{12}}\left(p_{1}\frac{a_{1}}{b_{1}}+p_{2}\frac{a_{2}}{b_{2}}-p_{4}\right)$ and $d_{5}^{12}=p_{1}\frac{c_{1}}{b_{1}}+p_{2}\frac{c_{2}}{b_{2}}$, if $p_{1}\frac{a_{1}}{b_{1}}+p_{2}\frac{a_{2}}{b_{2}}>p_{4}$, then $E_{5}^{12}\in \Sigma_{+}$, and at this equilibrium point, no Hopf bifurcation is possible because all eigenvalues are real. Due to the different signs of the eigenvalues, $E_{5}^{12}$ is a saddle point, and $x_{4}\left(t\right)$ may escape to infinity, which means that the virus may win. For $E_{5}^{13}$ and $E_{5}^{23}$, the results are similar.

**Interpretation for the interior equilibrium point** $E_{6}$.

For the equilibrium point $E_{6}\left(x_{1},x_{2},x_{3},v_{6}\right)$, where $x_{1}=$
$\frac{1}{b_{1}}\left(a_{1}-c_{1}v_{6}\right)$, $x_{2}=$
$\frac{1}{b_{2}}\left(a_{2}-c_{2}v_{6}\right)$, $x_{3}=$
$\frac{1}{b_{3}}\left(a_{3}-c_{3}v_{6}\right)$, and $v_{6}=\frac{1}{d_{6}}\left(p_{1}\frac{a_{1}}{b_{1}}+p_{2}\frac{a_{2}}{b_{2}}+p_{3}\frac{a_{3}}{b_{3}}-p_{4}\right)$, with $d_{6}=$
$p_{1}\frac{c_{1}}{b_{1}}+p_{2}\frac{c_{2}}{b_{2}}+$
$p_{3}\frac{c_{3}}{b_{3}}$, this equilibrium point is always a saddle point, and $x_{4}\left(t\right)$ may escape to infinity, which means that the virus may also win in this case. Since $E_{6}\in \Sigma_{+}$ if and only if $\frac{a_{i}}{c_{i}}>v_{6}>0$ for all i=1,2,3, we see that $x_{i}<\frac{a_{i}}{b_{i}}$ for all i=1,2,3 and $p_{1}\frac{a_{1}}{b_{1}}+p_{2}\frac{a_{2}}{b_{2}}+p_{3}\frac{a_{3}}{b_{3}}>p_{4}$. Then, we can conclude that if the levels of all three immunities (innate, humoral, and cellular immunity) decrease to lower levels than their normal levels, then the virus may win even though the immune system kills the virus at a rate higher than the rate of the virus’s proliferation. This case includes the possibility that the virus and the three immunities increase in number at the same time, but the immune system does not have the ability to eliminate the virus’s proliferation. Therefore, the quantity of virus $x_{4}\left(t\right)$ may escape to infinity if an orbit x(t) starts at a point $x_{0}\in \Sigma_{+}$ close to $E_{6}$, but $x_{0}\notin W_{E_{6}}^{s}$ or the quantity of virus $x_{4}\left(t\right)$ will converge to $v_{6}$ when *t* is large if $x_{0}\in W_{E_{6}}^{s}$. A stable limit cycle around $E_{6}$ cannot arise via a Hopf bifurcation since all eigenvalues are real.

In conclusion, after this analysis, we can state that there are sufficient conditions to completely destroy the virus only in a single case, namely, when $E_{3}\in \Sigma_{+}$ and $p_{4}<p_{1}\frac{a_{1}}{b_{1}}+p_{2}\frac{a_{2}}{b_{2}}+p_{3}\frac{a_{3}}{b_{3}}$, which means that $E_{3}$ is an attractive equilibrium point. In all other cases, the virus may win the fight against the human body’s immune system or, at best, the immune system may manage to limit the proliferation of the virus.

## 5. The Four-Dimensional Model with Two Types of Medical Control

So far, in our study, for the four-dimensional model introduced above, we have studied the interaction between a virus and the body’s immune system by considering only natural developments without external influences, such as drug administration or additional means of increasing one of the components of the immunity system. In this section, we develop a deterministic mathematical model for the case in which the interaction also depends on external factors, such as a good lifestyle, drug or vitamin administration, or even vaccination for increasing at least one of the types of immunity. In this sense, we choose suitable control functions for modeling this external influence by using the product and the sum between the variables, according to [6].

Similar studies were also carried out in [13,14,15,16] using different types of so-called control functions in order to obtain a suitable control for the modeled systems with two and three variables.

### 5.1. First Medical Control

In order to obtain a controlled pattern using external influences, we use three control functions—one in each of the three equations that model the behavior of the type of immunity—because an external intervention can usually be performed to strengthen the immune system but not for destroying or weakening the virus (see the last equation).

Therefore, by taking into account system (Equation 1), we propose the following four-dimensional first-order differential system:(4)$$\left\{\begin{array}{ccc} {\dot{x}}_{1} & = & a_{1}x_{1}-b_{1}x_{1}^{2}-c_{1}x_{1}x_{4}+\alpha x_{1}x_{2}x_{3}\\ {\dot{x}}_{2} & = & a_{2}x_{2}-b_{2}x_{2}^{2}-c_{2}x_{2}x_{4}+\beta x_{1}x_{2}x_{3}\\ {\dot{x}}_{3} & = & a_{3}x_{3}-b_{3}x_{3}^{2}-c_{3}x_{3}x_{4}+\gamma x_{1}x_{2}x_{3}\\ {\dot{x}}_{4} & = & p_{4}x_{4}-p_{1}x_{1}x_{4}-p_{2}x_{2}x_{4}-p_{3}x_{3}x_{4} \end{array}\right.$$
where α, β, and γ are real constants, and they are only used to improve the role of the immune system in the fight against the virus.

The fact that this controlled system also has 15 equilibrium points is very interesting, similar to the four-dimensional system without controls. Moreover, only $E_{3}$ and $E_{6}$ have other co-ordinates, and they are also very complicated to find. The rest of the 13 equilibrium points have the same co-ordinates, which is similar to the uncontrolled system (Equation 1). However, the local behavior of the system around these equilibrium points is very different compared to the system without controls (Equation 1).

The Jacobi matrix at an equilibrium point $\left(x_{1},x_{2},x_{3},x_{4}\right)$ is
$$\left(\begin{array}{cccc} a_{1}-2b_{1}x_{1}-c_{1}x_{4}+\alpha x_{2}x_{3} & \alpha x_{1}x_{3}\; & \alpha x_{1}x_{2} & -c_{1}x_{1}\\ \beta x_{2}x_{3} & a_{2}-2b_{2}x_{2}-c_{2}x_{4}+\beta x_{1}x_{3} & \beta x_{1}x_{2} & -c_{2}x_{2}\\ \gamma x_{2}x_{3} & \gamma x_{1}x_{3} & a_{3}-2b_{3}x_{3}-c_{3}x_{4}+\gamma x_{1}x_{2} & -c_{3}x_{3}\\ -p_{1}x_{4} & -p_{2}x_{4} & -p_{3}x_{4} & p_{4}-\sum_{i=1}^{3}p_{i}x_{i} \end{array}\right)$$

In order to find the equilibrium points by analyzing
(5)$$\left\{\begin{array}{c} x_{1}\left(a_{1}-b_{1}x_{1}-c_{1}x_{4}+\alpha x_{2}x_{3}\right)=0\\ x_{2}\left(a_{2}-b_{2}x_{2}-c_{2}x_{4}+\beta x_{1}x_{3}\right)=0\\ x_{3}\left(a_{3}-b_{3}x_{3}-c_{3}x_{4}+\gamma x_{1}x_{2}\right)=0\\ x_{4}\left(p_{4}-p_{1}x_{1}-p_{2}x_{2}-p_{3}x_{3}\right)=0 \end{array}\right.$$
we obtain the following equilibrium points:$$\begin{array}{cccc} E_{0}\left(0,0,0,0\right), & E_{1}^{1}\left(\frac{a_{1}}{b_{1}},0,0,0\right), & E_{1}^{2}\left(0,\frac{a_{2}}{b_{2}},0,0\right), & E_{1}^{3}\left(0,0,\frac{a_{3}}{b_{3}},0\right), \end{array}$$
$$\begin{array}{cccc} E_{2}^{12}\left(\frac{a_{1}}{b_{1}},\frac{a_{2}}{b_{2}},0,0\right), & E_{2}^{13}\left(\frac{a_{1}}{b_{1}},0,\frac{a_{3}}{b_{3}},0\right), & E_{2}^{23}\left(0,\frac{a_{2}}{b_{2}},\frac{a_{3}}{b_{3}},0\right), & E_{3}\left(x_{1},x_{2},x_{3},0\right), \end{array}$$
where $\left(x_{1},x_{2},x_{3},x_{4}\right)$ is a solution to (Equation 5) with $x_{i}>0$ for $i=\bar{1,3}$ and $x_{4}=0$.

Moreover, we have the following equilibrium points:$$E_{4}^{1}\left(\frac{p_{4}}{p_{1}},0,0,\frac{1}{c_{1}}\left(a_{1}-b_{1}\frac{p_{4}}{p_{1}}\right)\right),\;E_{4}^{2}\left(0,\frac{p_{4}}{p_{2}},0,\frac{1}{c_{2}}\left(a_{2}-b_{2}\frac{p_{4}}{p_{2}}\right)\right),\;E_{4}^{3}\left(0,0,\frac{p_{4}}{p_{3}},\frac{1}{c_{3}}\left(a_{3}-b_{3}\frac{p_{4}}{p3}\right)\right),$$
$$E_{5}^{12}\left(\frac{1}{b_{1}}\left(a_{1}-c_{1}v_{5}^{12}\right),\frac{1}{b_{2}}\left(a_{2}-c_{2}v_{5}^{12}\right),0,v_{5}^{12}\right),\;v_{5}^{12}=\frac{1}{d_{5}^{12}}\left(p_{1}\frac{a_{1}}{b_{1}}+p_{2}\frac{a_{2}}{b_{2}}-p_{4}\right),d_{5}^{12}=p_{1}\frac{c_{1}}{b_{1}}+p_{2}\frac{c_{2}}{b_{2}},$$
$$E_{5}^{13}\left(\frac{1}{b_{1}}\left(a_{1}-c_{1}v_{5}^{13}\right),0,\frac{1}{b_{3}}\left(a_{3}-c_{3}v_{5}^{13}\right),v_{5}^{13}\right),\;v_{5}^{13}=\frac{1}{d_{5}^{13}}\left(p_{1}\frac{a_{1}}{b_{1}}+p_{3}\frac{a_{3}}{b_{3}}-p_{4}\right),d_{5}^{13}=p_{1}\frac{c_{1}}{b_{1}}+p_{3}\frac{c_{3}}{b_{3}},$$
$$E_{5}^{23}\left(0,\frac{1}{b_{2}}\left(a_{2}-c_{2}v_{5}^{23}\right),\frac{1}{b_{3}}\left(a_{3}-c_{3}v_{5}^{23}\right),v_{5}^{23}\right),\;v_{5}^{23}=\frac{1}{d_{5}^{23}}\left(p_{2}\frac{a_{2}}{b_{2}}+p_{3}\frac{a_{3}}{b_{3}}-p_{4}\right),d_{5}^{23}=p_{2}\frac{c_{2}}{b_{2}}+p_{3}\frac{c_{3}}{b_{3}},$$
and $E_{6}\left(x_{1},x_{2},x_{3},x_{4}\right)$, where $\left(x_{1},x_{2},x_{3},x_{4}\right)$ is a solution to (Equation 5) with respect to $x_{i}>0$, $i=\bar{1,4}$.

For $E_{0}\left(0,0,0,0\right)$, we have the Jacobian $A=\left(\begin{array}{cccc} a_{1} & 0 & 0 & 0\\ 0 & a_{2} & 0 & 0\\ 0 & 0 & a_{3} & 0\\ 0 & 0 & 0 & p_{4} \end{array}\right)$ with eigenvalues $a_{1}$, $a_{2}$, $a_{3}$, and $p_{4}$; then, the equilibrium point at the origin is a repeller, which is an unstable node.

For $E_{1}^{1}\left(\frac{a_{1}}{b_{1}},0,0,0\right)$, we have the Jacobian $A=\left(\begin{array}{cccc} -a_{1} & 0 & 0 & -c_{1}\frac{a_{1}}{b_{1}}\\ 0 & a_{2} & 0 & 0\\ 0 & 0 & a_{3} & 0\\ 0 & 0 & 0 & p_{4}-p_{1}\frac{a_{1}}{b_{1}} \end{array}\right)$ with eigenvalues $-a_{1}$, $a_{2}$, $a_{3}$, and $\frac{p_{4}b_{1}-p_{1}a_{1}}{b_{1}}=p_{4}-p_{1}\frac{a_{1}}{b_{1}}$. Then, $E_{1}^{1}$ is a saddle point in the case of an uncontrolled system. The same situation will occur for $E_{1}^{2}$ and $E_{1}^{3}.$

For $E_{2}^{12}\left(\frac{a_{1}}{b_{1}},\frac{a_{2}}{b_{2}},0,0\right)$, we have $A=\left(\begin{array}{cccc} -a_{1} & 0 & \alpha \frac{a_{1}}{b_{1}}\frac{a_{2}}{b_{2}} & -c_{1}\frac{a_{1}}{b_{1}}\\ 0 & -a_{2} & \beta \frac{a_{1}}{b_{1}}\frac{a_{2}}{b_{2}} & -c_{2}\frac{a_{2}}{b_{2}}\\ 0 & 0 & a_{3}+\gamma \frac{a_{1}}{b_{1}}\frac{a_{2}}{b_{2}} & 0\\ 0 & 0 & 0 & p_{4}-p_{1}\frac{a_{1}}{b_{1}}-p_{2}\frac{a_{2}}{b_{2}} \end{array}\right)$, with the following eigenvalues: $-a_{1}$, $-a_{2}$, $\frac{a_{3}b_{1}b_{2}+\gamma a_{1}a_{2}}{b_{1}b_{2}}$, and $p_{4}-p_{1}\frac{a_{1}}{b_{1}}-p_{2}\frac{a_{2}}{b_{2}}$.

As a consequence, we obtain the following results:

**Theorem** **5.**
*$E_{2}^{12}$ is an attractor if and only if $p_{4}<p_{1}\frac{a_{1}}{b_{1}}+p_{2}\frac{a_{2}}{b_{2}}$ and $a_{3}b_{1}b_{2}+\gamma a_{1}a_{2}<0$. Otherwise, $E_{2}^{12}$ is a saddle point.*


In conclusion, despite the missing third type of immunity, if the level of the first two types of immunity is higher than the rate of viral proliferation ($p_{1}\frac{a_{1}}{b_{1}}+p_{2}\frac{a_{2}}{b_{2}}>p_{4}$) and $\gamma <-a_{3}\frac{b_{1}b_{2}}{a_{1}a_{2}}<0$, then the immune system may win the fight against the virus. This happens because any orbit starting at a point close to $E_{2}^{12}$ will converge to $E_{2}^{12}$ when t substantially increases. Thus, even though the initial level of cellular immunity is very low when the virus appears, the immune system may win the fight if innate immunity and humoral immunity increase beyond the normal threshold using medical interventions during the early stages of viral infection (namely, $\gamma <-a_{3}\frac{b_{1}b_{2}}{a_{1}a_{2}}<0$).

A similar scenario occurs for $E_{2}^{13}$ and $E_{2}^{23}$ for humoral immunity and innate immunity, respectively.

Due to complicated computations, for equilibrium points $E_{3}$, $E_{4}^{i}$ (i=1,2,3), $E_{5}^{ij}$ (1≤i<j≤3), and $E_{6}$, it is very difficult to obtain some analytical results and medical interpretations at this moment.

### 5.2. Second Medical Control

Next, if we use the other three control functions, we can consider the following four-dimensional first-order differential system:(6)$$\left\{\begin{array}{ccc} {\dot{x}}_{1} & = & a_{1}x_{1}-b_{1}x_{1}^{2}-c_{1}x_{1}x_{4}+\alpha x_{1}\left(x_{2}+x_{3}\right)\\ {\dot{x}}_{2} & = & a_{2}x_{2}-b_{2}x_{2}^{2}-c_{2}x_{2}x_{4}+\beta x_{2}\left(x_{3}+x_{1}\right)\\ {\dot{x}}_{3} & = & a_{3}x_{3}-b_{3}x_{3}^{2}-c_{3}x_{3}x_{4}+\gamma x_{3}\left(x_{1}+x_{2}\right)\\ {\dot{x}}_{4} & = & p_{4}x_{4}-p_{1}x_{1}x_{4}-p_{2}x_{2}x_{4}-p_{3}x_{3}x_{4} \end{array}\right.$$
where α, β, and γ are also real constants, and they are only used to improve the role of the immune system in the fight against the virus.

The Jacobi matrix at an equilibrium point $\left(x_{1},x_{2},x_{3},x_{4}\right)$ is
$$\left(\begin{array}{cccc} a_{1}-2b_{1}x_{1}-c_{1}x_{4}+\alpha x_{2}+\alpha x_{3} & \alpha x_{1} & \alpha x_{1} & -c_{1}x_{1}\\ \beta x_{2} & a_{2}-2b_{2}x_{2}-c_{2}x_{4}+\beta x_{3}+\beta x_{1}\; & \beta x_{2} & -c_{2}x_{2}\\ \gamma x_{3} & \gamma x_{3} & a_{3}-2b_{3}x_{3}-c_{3}x_{4}+\gamma x_{1}+\gamma x_{2} & -c_{3}x_{3}\\ -p_{1}x_{4} & -p_{2}x_{4} & -p_{3}x_{4} & p_{4}-\sum_{i=1}^{3}p_{i}x_{i} \end{array}\right)$$

We analyze the following nonlinear system with four unknowns:(7)$$\left\{\begin{array}{c} x_{1}\left(a_{1}-b_{1}x_{1}-c_{1}x_{4}+\alpha x_{2}+\alpha x_{3}\right)=0\\ x_{2}\left(a_{2}-b_{2}x_{2}-c_{2}x_{4}+\beta x_{1}+\beta x_{3}\right)=0\\ x_{3}\left(a_{3}-b_{3}x_{3}-c_{3}x_{4}+\gamma x_{1}+\gamma x_{2}\right)=0\\ x_{4}\left(p_{4}-p_{1}x_{1}-p_{2}x_{2}-p_{3}x_{3}\right)=0 \end{array}\right.$$
and we immediately obtain the following equilibrium points:$$\begin{array}{cccc} E_{0}\left(0,0,0,0\right), & E_{1}^{1}\left(\frac{a_{1}}{b_{1}},0,0,0\right), & E_{1}^{2}\left(0,\frac{a_{2}}{b_{2}},0,0\right), & E_{1}^{3}\left(0,0,\frac{a_{3}}{b_{3}},0\right). \end{array}$$

Moreover, equilibrium points $E_{2}^{12}\left(x_{1},x_{2},0,0\right)$, $E_{2}^{13}\left(x_{1},0,x_{3},0\right)$, and $E_{2}^{23}\left(0,x_{2},x_{3},0\right)$ can exist, with positive co-ordinates, only under certain conditions that system (Equation 7) must meet. The equilibrium point $E_{3}\left(x_{1},x_{2},x_{3},0\right)$ is a unique equilibrium with respect to $x_{4}=0$ if and only if the determinant of matrix $\left(\begin{array}{ccc} b_{1} & -\alpha & -\alpha \\ -\beta & b_{2} & -\beta \\ -\gamma & -\gamma & b_{3} \end{array}\right)$ is not equal to 0 (i.e., $b_{1}b_{2}b_{3}-2\alpha \beta \gamma -b_{1}\beta \gamma -b_{2}\alpha \gamma -b_{3}\alpha \beta \neq 0$) and $x_{i}>0$, i=1,2,3. Otherwise, we can obtain an infinity of equilibrium points of this type or none at all.

Due to complicated computations, which is the case of the first controlled system (Equation 4), a study of the behavior of this second controlled system with respect to the other equilibrium points is very hard to carry out. The study is related to the following equilibrium points:$$\begin{array}{ccc} E_{4}^{1}\left(\frac{p_{4}}{p_{1}},0,0,\frac{1}{c_{1}}\left(a_{1}-b_{1}\frac{p_{4}}{p_{1}}\right)\right), & E_{4}^{2}\left(0,\frac{p_{4}}{p_{2}},0,\frac{1}{c_{2}}\left(a_{2}-b_{2}\frac{p_{4}}{p_{2}}\right)\right), & E_{4}^{3}\left(0,0,\frac{p_{4}}{p_{3}},\frac{1}{c_{3}}\left(a_{3}-b_{3}\frac{p_{4}}{p_{3}}\right)\right), \end{array}$$
where $E_{5}^{ij}\left(x_{1},x_{2},x_{3},x_{4}\right)$, with $x_{i}>0$, $x_{j}>0$, 1≤i<j≤3, $x_{k}=0$, k∈{1,2,3}∖{i,j}, $x_{4}>0$, and finally, $E_{6}\left(x_{1},x_{2},x_{3},x_{4}\right)$, where $x_{i}>0$, i=1,2,3,4.

For $E_{0}\left(0,0,0,0\right)$, we have the Jacobian $A=\left(\begin{array}{cccc} a_{1} & 0 & 0 & 0\\ 0 & a_{2} & 0 & 0\\ 0 & 0 & a_{3} & 0\\ 0 & 0 & 0 & p_{4} \end{array}\right)$ with eigenvalues $a_{1}$, $a_{2}$, $a_{3}$, and $p_{4}$, and the equilibrium point at the origin is a repeller, which is an unstable node.

For $E_{1}^{1}\left(\frac{a_{1}}{b_{1}},0,0,0\right)$, we have the Jacobian $A=\left(\begin{array}{cccc} -a_{1} & \alpha \frac{a_{1}}{b_{1}} & \alpha \frac{a_{1}}{b_{1}} & -c_{1}\frac{a_{1}}{b_{1}}\\ 0 & a_{2}+\beta \frac{a_{1}}{b_{1}} & 0 & 0\\ 0 & 0 & a_{3}+\gamma \frac{a_{1}}{b_{1}} & 0\\ 0 & 0 & 0 & p_{4}-p_{1}\frac{a_{1}}{b_{1}} \end{array}\right)$ with eigenvalues $-a_{1}$, $\frac{a_{2}b_{1}+\beta a_{1}}{b_{1}}$, $\frac{a_{3}b_{1}+\gamma a_{1}}{b_{1}}$, and $p_{4}-p_{1}\frac{a_{1}}{b_{1}}$.

Then, we have the following result.

**Theorem** **6.**
*$E_{1}^{1}$ is an attractor if and only if $p_{4}<p_{1}\frac{a_{1}}{b_{1}}$ and $a_{2}b_{1}+\beta a_{1}<0$, $a_{3}b_{1}+\gamma a_{1}<0$.*

*Otherwise, $E_{1}^{1}$ is a saddle point.*


In conclusion, despite the missing second and third types of immunity (humoral immunity and cellular immunity), if the level of the first type of immunity (innate immunity) is higher than the rate of viral proliferation ($p_{1}\frac{a_{1}}{b_{1}}>p_{4}$) and $\beta <-a_{2}\frac{b_{1}}{a_{1}}<0$ and $\gamma <-a_{3}\frac{b_{1}}{a_{1}}<0$, then the immune system may win the fight against the virus. This happens because any orbit starting at a point close to $E_{1}^{1}$ will converge to $E_{1}^{1}$ when t substantially increases. Therefore, even though the initial level of the last two categories of immunity is very low when the virus appears, the immune system may win the fight if the level of the first immunity category increases beyond the normal threshold using medical interventions in the early stages of viral infection (i.e., $\beta <-a_{2}\frac{b_{1}}{a_{1}}<0$, $\gamma <-a_{3}\frac{b_{1}}{a_{1}}<0$).

A similar scenario occurs for $E_{1}^{2}$ and $E_{1}^{3}$ with respect to humoral immunity and cellular immunity, respectively. Indeed, for $E_{1}^{2}\left(0,\frac{a_{2}}{b_{2}},0,0\right)$, we have the Jacobian
$$A=\left(\begin{array}{cccc} a_{1}+\alpha \frac{a_{2}}{b_{2}} & 0 & 0 & 0\\ \beta \frac{a_{2}}{b_{2}} & -a_{2} & \beta \frac{a_{2}}{b_{2}} & -c_{2}\frac{a_{2}}{b_{2}}\\ 0 & 0 & a_{3}+\gamma \frac{a_{2}}{b_{2}} & 0\\ 0 & 0 & 0 & p_{4}-p_{2}\frac{a_{2}}{b_{2}} \end{array}\right)$$
with eigenvalues $\frac{a_{1}b_{2}+\alpha a_{2}}{b_{2}}$, $-a_{2}$, $\frac{a_{3}b_{2}+a_{2}\gamma }{b_{2}}$, and $p_{4}-p_{2}\frac{a_{2}}{b_{2}}$; moreover, for $E_{1}^{3}\left(0,0,\frac{a_{3}}{b_{3}},0\right)$, we have the Jacobian
$$A=\left(\begin{array}{cccc} a_{1}+\alpha \frac{a_{3}}{b_{3}} & 0 & 0 & 0\\ 0 & a_{2}+\beta \frac{a_{3}}{b_{3}} & 0 & 0\\ \gamma \frac{a_{3}}{b_{3}} & \gamma \frac{a_{3}}{b_{3}} & -a_{3} & -c_{3}\frac{a_{3}}{b_{3}}\\ 0 & 0 & 0 & p_{4}-p_{3}\frac{a_{3}}{b_{3}} \end{array}\right)$$
with eigenvalues $\frac{a_{1}b_{3}+\alpha a_{3}}{b_{3}}$, $\frac{a_{2}b_{3}+\beta a_{3}}{b_{3}}$, and $-a_{3}$, $p_{4}-p_{3}\frac{a_{3}}{b_{3}}$.

There is no equilibrium point $E_{2}^{12}\left(\frac{a_{1}b_{2}+a_{2}\alpha }{b_{1}b_{2}-\alpha \beta },\frac{a_{2}b_{1}+a_{1}\beta }{b_{1}b_{2}-\alpha \beta },0,0\right)$ if the determinant of matrix $\left(\begin{array}{ccc} b_{1} & -\alpha & a_{1}\\ -\beta & b_{2} & a_{2}\\ -\gamma & -\gamma & a_{3} \end{array}\right)$ is non-zero. However, if equilibrium $E_{2}^{12}$ exists, then the Jacobi matrix is too large to write here; however, we can write the characteristic polynomial as follows:$$\left(X-\left(a_{3}+\gamma \frac{a_{1}b_{2}+a_{2}\alpha }{b_{1}b_{2}-\alpha \beta }+\gamma \frac{a_{2}b_{1}+a_{1}\beta }{b_{1}b_{2}-\alpha \beta }\right)\right)\left(X-\left(p_{4}-p_{1}\frac{a_{1}b_{2}+a_{2}\alpha }{b_{1}b_{2}-\alpha \beta }-p_{2}\frac{a_{2}b_{1}+a_{1}\beta }{b_{1}b_{2}-\alpha \beta }\right)\right)$$
$$\left(X^{2}+\frac{b_{2}a_{2}b_{1}+b_{2}\beta a_{1}+\alpha a_{2}b_{1}+a_{1}b_{2}b_{1}}{b_{1}b_{2}-\alpha \beta }X+\frac{a_{1}b_{2}a_{2}b_{1}+\alpha a_{2}^{2}b_{1}+a_{1}^{2}b_{2}\beta +\beta a_{2}\alpha a_{1}}{b_{1}b_{2}-\alpha \beta }\right)$$.

Then, equilibrium point $E_{2}^{12}$ is an attractor if and only if $a_{3}+\gamma \frac{a_{1}b_{2}+a_{2}\alpha }{b_{1}b_{2}-\alpha \beta }+\gamma \frac{a_{2}b_{1}+a_{1}\beta }{b_{1}b_{2}-\alpha \beta }<0$, $p_{4}-p_{1}\frac{a_{1}b_{2}+a_{2}\alpha }{b_{1}b_{2}-\alpha \beta }-p_{2}\frac{a_{2}b_{1}+a_{1}\beta }{b_{1}b_{2}-\alpha \beta }<0$, $\frac{b_{1}\left(a_{1}b_{2}+a_{2}\alpha \right)+b_{2}\left(a_{2}b_{1}+a_{1}\beta \right)}{b_{1}b_{2}-\alpha \beta }=b_{1}\frac{a_{1}b_{2}+a_{2}\alpha }{b_{1}b_{2}-\alpha \beta }+b_{2}\frac{a_{2}b_{1}+a_{1}\beta }{b_{1}b_{2}-\alpha \beta }>0$, and $\frac{\left(a_{1}b_{2}+a_{2}\alpha \right)\left(a_{2}b_{1}+a_{1}\beta \right)}{b_{1}b_{2}-\alpha \beta }>0$.

As a consequence, we obtain the following result.

**Theorem** **7.**
*If $E_{2}^{12}$ exists, then the equilibrium point $E_{2}^{12}$ is an attractor if and only if*

$$\begin{array}{c} p_{4}<p_{1}\frac{a_{1}b_{2}+a_{2}\alpha }{b_{1}b_{2}-\alpha \beta }+p_{2}\frac{a_{2}b_{1}+a_{1}\beta }{b_{1}b_{2}-\alpha \beta }\;,\;\;\gamma <-a_{3}{\left(\frac{a_{1}b_{2}+a_{2}\alpha }{b_{1}b_{2}-\alpha \beta }+\frac{a_{2}b_{1}+a_{1}\beta }{b_{1}b_{2}-\alpha \beta }\right)}^{-1}\\ a_{1}b_{2}+\alpha a_{2}>0\;,\;\;a_{2}b_{1}+\beta a_{1}>0. \end{array}$$


*Otherwise, $E_{2}^{12}$ is a saddle point.*


Thus, despite the missing third type of immunity (cellular immunity), if the level of the first two types of immunity (innate immunity and humoral immunity) is sufficient such that the rate of viral proliferation $p_{4}$ satisfies condition $p_{4}<p_{1}\frac{a_{1}b_{2}+a_{2}\alpha }{b_{1}b_{2}-\alpha \beta }+p_{2}\frac{a_{2}b_{1}+a_{1}\beta }{b_{1}b_{2}-\alpha \beta }$ and the level of the first two immunity categories can increase beyond the normal thresholds using medical interventions, namely $\alpha >-a_{1}\frac{b_{2}}{a_{2}}$, $\beta >-a_{2}\frac{b_{1}}{a_{1}}$ and $\gamma <-a_{3}{\left(\frac{a_{1}b_{2}+a_{2}\alpha }{b_{1}b_{2}-\alpha \beta }+\frac{a_{2}b_{1}+a_{1}\beta }{b_{1}b_{2}-\alpha \beta }\right)}^{-1}<0$, then the immune system may win the fight against the virus. This happens because any orbit starting at a point close to $E_{2}^{12}$ will converge to $E_{2}^{12}$ when t is very large and whenever $E_{2}^{12}$ is an attractor.

Moreover, by using this second medical control model (Equation 6), the immune system may win the fight even if the levels of the first two immunity types (innate and humoral) are not at the maximum threshold values of $\frac{a_{i}}{b_{i}}$, i=1,2. In order to clarify this, we observe that the co-ordinates of equilibrium point $E_{2}^{12}$. If $E_{2}^{12}$ exists; thus, it is possible to obtain $\frac{a_{1}b_{2}+a_{2}\alpha }{b_{1}b_{2}-\alpha \beta }<\frac{a_{1}}{b_{1}}$ and $\frac{a_{2}b_{1}+a_{1}\beta }{b_{1}b_{2}-\alpha \beta }<\frac{a_{2}}{b_{2}}$ if $a_{1}b_{2}+\alpha a_{2}<0$, $a_{2}b_{1}+\beta a_{1}<0$, and $b_{1}b_{2}-\alpha \beta <0$. However, in this case, $E_{2}^{12}$ is a saddle point; then, the elimination of the virus will be possible only if orbit x(t) of the system (Equation 6) begins from point $x_{0}\in \Sigma_{+}$, which is very close to $E_{2}^{12}$, and this starting point $x_{0}$ belongs to the local stable manifold $W_{E_{2}^{12}}^{s}$.

In conclusion, even when the initial level of the last category of immunity is very low when the virus appears, the immune system may win the fight against the virus if the level of the first two immunity categories can increase using external interventions in the early stages of infection.

A similar scenario occurs for $E_{2}^{13}$ and $E_{2}^{23}$ with respect to humoral immunity and innate immunity, respectively.

Due to complicated computations, for equilibrium points $E_{3}$, $E_{4}^{i}$ (i=1,2,3), $E_{5}^{ij}$ (1≤i<j≤3), and $E_{6}$, obtaining the analytical results and medical interpretations is very difficult at this moment.

## 6. Interpretations, Limitations, and Future Research

In this manuscript, we have presented a study about the interplay between the human body’s immune system and a pathogenic agent (virus, microbe, bacteria, parasites, or fungi). We have introduced a deterministic mathematical pattern based on a system of first-order differential equations in order to model this interaction using the tools of dynamical systems theory, and the pattern is used to analyze the local dynamics of the obtained dynamical system. Because the proposed model is not based on clinical studies or other empirical data, the approach is exclusively a qualitative theoretical study and reveals the importance of all three immunity types (innate, humoral, and cellular) in the fight against any pathogenic agent.

Moreover, in this study, we have introduced two types of interaction patterns with controls, and we have also obtained several very interesting theoretical medical interpretations for these controlled systems. Although they are not verified by empirical data, among these interpretations, some useful assertions arise from this study:

1. If the immune system is sufficiently weak when the virus begins its multiplication, then the virus has a better chance of defeating the immune system of the body.

2. A deficiency in the quantity of a single type of immunity at the beginning of the virus’s proliferation may lead to the loss of the fight against the virus.

3. During the infection, if the immunity levels decrease considerably compared to their normal concentrations, then the pathogen agent may win even if the immune system destroys the virus at a higher rate than the rate of the virus’s multiplication.

4. If the immunity levels are within their normal concentrations from the beginning of the infection and the immune system is in a good position to destroy the virus at a high rate, then the immune system has a better chance of winning the fight against the virus.

5. If the concentration of at least one type of immunity can increase beyond its normal threshold using medical interventions at the beginning of the infection, then the immune system has a better chance of eliminating the virus.

This last assertion is extremely important because it reveals the possibility of winning the fight against a virus by increasing the level of at least one immunity category. In our opinion, this natural idea should be further exploited in medical studies.

The results obtained in this study and also in a recent paper [28] suggest that the scientific community should develop new methods for reinforcing the immune system in order to fight against pathogenic agents in an improved manner.

Maybe one of the first mathematical models for the growth of populations was introduced by the famous Thomas Robert Malthus, theoretician and economist (1766–1834), representing an exponential model with inevitable population growth due to the improvement of people’s life conditions [29]. However, this model by Malthus was refuted by Belgian mathematician Pierre François Verhulst (1804–1849) by introducing a new mathematical model based on the logistic function and the carrying capacity (as an upper threshold) [30,31,32]. Moreover, in 1825, in order to obtain a law of mortality for populations, British mathematician Benjamin Gompertz introduced another demographic model based on the concept of carrying capacity, but it used the Gompertz logarithmic function [33]. After 140 years, in 1964, A.K. Laird used the Gompertz curve to fit data on the growth of tumors [34]. Moreover, in order to model many growth phenomena from oncology and epidemiology, FJ Richards proposed a generalization of the logistic function in 1959 [35]. Additionally, each Kermack-McKendrick or Lotka-Volterra deterministic model for the interaction between two or more species is based on the Verhulst logistic function with the corresponding carrying capacity of each species [36,37,38].

All these theoretical models were considered valid for mathematical research and represented the basics for the mathematical modeling of the phenomena from biology, ecology, epidemiology, and sociology. A lot of generalizations of these models have been introduced and studied by several authors, including the papers [6,7,12,13,14,15,16,17,28].

When taking into account the very complex relationships between the components of the immune system of the human body, and since empirical proof of the model cannot be provided at this stage, we can conclude that the current study is purely theoretical. However, by taking into account the historical context and tradition of these models, it follows that the proposed models and the used methods have the potential to be useful for practical application in the future. The patterns introduced and studied in this work, with or without control functions, are based on Verhulst’s logistic model [30,31,32] and the classical Lotka–Volterra predator–prey model [36,37,38]. These models are widely accepted in the scientific world and have been confirmed by theoretical and practical research carried out in recent years. In this sense, we present the following: in the paper [39], experimental conditions were designed in order to use the Lotka–Volterra model for tumor cell line interactions; in [40], quantitative models were integrated with experimental and clinical data in order to study the clonal interactions in cancer; in [41], it was successfully tested whether a Lotka–Volterra model is appropriate for describing microbial interactions by using a set of simple in vitro experiments; in [42], structural identifiability analyses were used in order to find the extent to which a time series of relative abundances can be used to parameterize the generalized Lotka–Volterra model.

Furthermore, a mathematical model for optimal social contracts based on a first-order differential equation was studied in [43], and an approach to the spatial scale effects in epidemic spread models was undertaken in [44]. Moreover, an improved Lotka–Volterra pattern using quantum game theory was investigated in [45], and the classical Lotka-Volterra model was used in order to obtain a global output tracking control by an input-to-state stability analysis of the closed-loop systems [46].

A possible next piece of work may address a computational study of this model in order to present some numerical simulations. Moreover, the approach can be significantly improved by proposing practical empirical applications using data.

## Data Availability

Not applicable.

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
