# Peer review of "A Local Analysis of a Mathematical Pattern for Interactions between the Human Immune System and a Pathogenic Agent"

_entropy, 2023, doi:10.3390/e25101392_

Round 1

Reviewer 1 Report

The authors presented a ODE model for the interaction between  human immune system and virus. However, in my opinion, there should be some biological evidence to support their research, which I didn't find in this manuscript. Therefore, I don't recommend it for publication.

Author Response

Response to Reviewer 1 Comments

Dear reviewer,

Thank you very much for your comments!

The Reviewer 1 comments are:

"The authors presented a ODE model for the interaction between  human immune system and virus. However, in my opinion, there should be some biological evidence to support their research, which I didn't find in this manuscript. Therefore, I don't recommend it for publication."

Response:

My work is just a qualitative theoretical and mathematical study of the interaction between the immune human system and a pathogenic agent by a deterministic mathematical four dimensional pattern. This deterministic model represented by the system with four first order ordinary differential equations (1) is obtained by using the three assumptions A1-A3. These hypothesis are inspired from classical logistic model of Verhulst (1847) and from prey-predator Lotka-Volterra (1910-1920) type models. Although, these are only theoretical patterns, they was confirmed by the realities from biology, epidemiology or medicine.

Thank you again for all suggestions, remarks and comments!

Reviewer 2 Report

The present manuscript is original, very interesting and important from a theoretical perspective, but lacks empirical testing of the proposed model with real data.

In this sense, the proposed conclusions should be rather considered as hypotheses and not conclusions, since they still require testing.

Author Response

Response to Reviewer 2 Comments

Dear Reviewer,
Thank you very much for your suggestions, remarks and comments!

The Reviewer 2 comments are:

"The present manuscript is original, very interesting and important from a theoretical perspective, but lacks empirical testing of the proposed model with real data.

In this sense, the proposed conclusions should be rather considered as hypotheses and not conclusions, since they still require testing."

Author's response:

My work is only a qualitative theoretical study of the interaction between the immune human system and a pathogenic agent by a deterministic mathematical four dimensional pattern. I have considered the interactions between the three types of immunities (innate, humoral and cellular). Moreover, in the fifth section I have introduced two types of models with control and I obtained a lot of very interesting medical interpretations.  This deterministic model represented by the system with four first order ordinary differential equations (1) form page 4 is obtained by using the three hypothesis A1-A3 from the second section. These hypothesis are inspired from classical logistic model of Verhulst (1847) and from prey-predator Lotka-Volterra (1910-1920) type models.

The proposed model is deterministic, not stochastic or probabilistic. I used the formalism of dynamical systems defined by a system of first order differential equations. The entire study is only local, by using the famous and classical Hartman-Grobman Theorem. That means that we are interested to obtain the behavior of the associated dynamical system in a neighborhood of each equilibrium point. The motivation is to study this mathematical model in order to obtain the local behavior and so, to conclude some medical interpretations.

I will try to do empirical tests of the proposed model with real data, but will be very hard for me (as Math teacher), because I have not access to the medical infrastructure.

I will precise that the obtained conclusions are only theoretic conclusions because they are not tested through real dates.

Thank you again for all suggestions, remarks and comments!

Reviewer 3 Report

Except for extensive editorial (English) corrections, I did not find any flaws in the manuscript. For example, Line 27 has two missing words. 

Additionally, the 'Introduction' section is too long and it should be shortened. Since this kind of article does not require results and discussion, the methodology and proofs of theorems should fill in for those items.

Extensive editorial assistance is needed.

Author Response

Response to Reviewer 3 Comments 

Dear Reviewer,

Thank you very much for your suggestions, remarks and comments!

The Reviewer 3 comments are:

"Except for extensive editorial (English) corrections, I did not find any flaws in the manuscript. For example, Line 27 has two missing words. 

Additionally, the 'Introduction' section is too long and it should be shortened. Since this kind of article does not require results and discussion, the methodology and proofs of theorems should fill in for those items."

Author's response:

Thank you very much for all your attention. I will make all the necessary corrections so that my article looks very good from the point of view of the English language, including the extensive editorial assistance for the improvement of the English of the manuscript.

I will change the introductory section in order to be more short and more concise. Thank you very much for this suggestion!

I will follow the advise to put the results and discussions  after the methodology and the proofs of theorems.. Thank you very much for this suggestion!

Thank you again for all suggestions, remarks and comments!

Round 2

Reviewer 1 Report

The author claimed in their response, "Although, these are only theoretical patterns, they WAS confirmed by the realities from biology, epidemiology or medicine." I suggest they added some related references that confirm their theoretical results before it can be accepted. 

Author Response

Response to Reviewer 1 Comments

Dear reviewer,

Thank you very much for your suggestions, remarks and comments!

The Reviewer 1 comments are:

"The author claimed in their response, "Although, these are only theoretical patterns, they WAS confirmed by the realities from biology, epidemiology or medicine." I suggest they added some related references that confirm their theoretical results before it can be accepted. "

Author's response:

Thank you very much for this suggestion and for all observations and comments which leaded to the improvement of my work!

I will add some related references that confirmed the theoretical results of the mathematical patterns applied in biology, epidemiology and medicine, like:

Martcheva, M. Texts in Applied Mathematics. In An Introduction to Mathematical Epidemiology; Springer: New York, NY, USA, 2015; Volume 61, pp. 33–66.

Maybe one of first mathematical model for the growth of populations was introduced by the famous Thomas Robert Malthus, theoretician and economist (1766-1834), by an exponential model with an inevitable population growth due to  the improvement of the people life conditions. See “An Essay on the Principle of Population As It Affects the Future Improvement of Society, with Remarks on the Speculations of Mr. Goodwin, M. Condorcet and Other Writers” (1 ed.). London: J. Johnson in St Paul's Church-yard. 1798. Retrieved 20 June 2015 via Internet Archive (https://archive.org/details/essayonprincipl00malt/page/n1/mode/2up). But this model of Malthus was refuted by Belgian mathematician Pierre François Verhulst (1804-1849) by introducing a new mathematical model based on the logistic function and the carrying capacity (as an upper threshold) in the following papers:

  1. Verhulst, P.F. Notice sur la loi que la population poursuit dans son accroissement. Corresp. Math. Phys. 1838, 10, 113–121.
  2. Verhulst, P.F. Recherches mathématiques sur la loi d’accroissement de la population. Nouv. Mém. l’Acad. R. Sci. Belles-Lettres Brux. 1845, 18, 8.
  3. Verhulst, P.F. Deuxième mémoire sur la loi d’accroissement de la population. Mém. l’Acad. R. Sci. Lettres Beaux-Arts Belg. 1847, 20, 1–32.

Also, in 1825, in order to obtain a law of mortality for populations, British mathematician Benjamin Gompertz introduced another demographic model based on the concept of carrying capacity, but using the Gompertz logarithmic function. One hundred forty years later in 1964, A.K. Laird used the Gompertz curve to fit data on the growth of tumors. Also, in order to model many growth phenomena from oncology and epidemiology, F. J. Richards proposed a generalization of the logistic function in 1959. See the papers:

  1. Gompertz, B. On the nature of the function expressive of the law of human mortality, and on a new mode of determining the value of life contingencies. Philos. Trans. R. Soc. Lond. 1825, 115, 513–585.
  2. Laird, A.K. Dynamics of Tumor Growth. Br. J. Cancer 1964, 13, 490–502.
  3. Richards, F.J. A Flexible Growth Function for Empirical Use. J. Exp. Bot. 1959, 10, 290–300.

Moreover, every Kermack-McKendrick or Lotka-Volterra deterministic models for the interaction between species is based on the Verhulst logistic function with the carrying capacity of each species. For a good understanding of all these model, it is very indicated the classics works:

  1. J. Lotka, Elements of Physical Biology (Williams-Wilkins, Baltimore, 1925) 460 pages.
  2. Volterra, Leçon sur la Theorie Mathématique de la lute pour la vie (Gauthier-Villars, Paris, 1931) 226 pages.
  3. Volterra, Principles de biologie mathématique, Acta Biother. 3, 1-36 (1937).

          or the recent book:

  1. Martcheva, M. Texts in Applied Mathematics. In An Introduction to Mathematical Epidemiology; Springer: New York, NY, USA, 2015; Volume 61, pp. 33–66.

All these theoretical model was considered valid for the mathematical researches and represent the basic for the mathematical modelling of the phenomena from biology, ecology, epidemiology. A lot of generalizations was introduced and studied by several authors, among which the papers [6], [12-17], [28] and the current manuscript.  

Thank you again! 

Reviewer 2 Report

The author has clarified the nature of the work presented, and revised the manuscript accordingly following my comments on the original version. I have no further issues.

Author Response

Response to Reviewer 2 Comments

             Dear Reviewer,

Thank you very much for your suggestions, remarks and comments!

The Reviewer 2 comments are:

"The author has clarified the nature of the work presented, and revised the manuscript accordingly following my comments on the original version. I have no further issues."

Author's response:

Dear Reviewer,
Thank you very much for all your attention, comments, remarks observations and recommendations which conducted to the improvement of my manuscript!